# Formalizing and Falsifying Causal Pathways of Rare Events

**Anahita Haghighat** [1]   **Dominik Janzing** [2]

## Abstract

Building on recent formalizations of root cause analysis for rare events ("outliers") in structural equation models, we propose a formal definition of a causal pathway and discuss its testable implications. We identify conditions under which these implications depend only on a causal abstraction defined by the pathway of rare events, rather than on the full causal graph of the underlying system. Accordingly, we introduce an abstraction of causal structure to pathways of rare events that bridges simple verbal causal explanations and detailed causal modeling.

## 1. Introduction

Causal explanations (Halpern & Pearl, 2005) are of particular interest when an observed event is unexpected under a given probabilistic model. Typical examples of rare events include natural disasters (Hannart et al., 2016), stock market crashes (Sanyal et al., 2025), failures in technical systems (Pham et al., 2024), and extreme gene expression patterns (Li et al., 2025). Current approaches to root cause analysis of rare events are typically centered on identifying a small set of causal origins, called *root causes* (Orchard et al., 2025; Nagalapatti et al., 2025; Lin et al., 2024; Li et al., 2022; Budhathoki et al., 2022a; Gnecco et al., 2021; Liu et al., 2021; Lin et al., 2018). While root cause analysis is informative, it does not capture the mechanisms through which these causes propagate to produce the observed outcome, which may include interacting mechanisms and modulated contextual factors. As a result, causal explanations cannot, in general, be reduced to identifying a limited number of root causes. For purposes of interpretability and scientific insight, it is therefore desirable to provide causal explanations that identify not only root causes, but also the relevant subgraph of the causal model that connects these causes to the observed event. Furthermore, the full causal model is often

[1]Independent [2]Amazon Research, Tübingen, Germany. Correspondence to: Dominik Janzing <janzind@amazon.com>.

*Proceedings of the 43rd International Conference on Machine Learning*, Seoul, South Korea. PMLR 306, 2026. Copyright 2026 by the author(s).

not known (Ikram et al., 2025), therefore causal discovery can also be viewed as the problem of identifying a minimal causal structure that strongly explains a particular observed event (Beckers, 2022). This perspective highlights the need for a formal notion of event-specific causal explanation that remains well-defined under partial model knowledge and uncertainty about the underlying detailed causal graph.

**Our contributions.** We introduce a formalization of *causal pathways* of events connecting root causes to a target. Based on this framework, we define several quantities that measure the quality of an explanation, motivated by the ideas of *plausibility* and *falsifiability*. Although most of our events refer to numeric variables, we emphasize that our concepts generalize to variables with arbitrary ranges, e.g., variables may describe tokens or sentences when a probability distribution is defined via semantic embedding.

**Related work.** Existing work on causal modeling under rare events has largely focused on **extreme-value regimes** with asymptotic or tail-based limits, often subject to parametric assumptions (Engelke et al., 2025; Klüppelberg & Krali, 2026). While this line of work provides insight into causal mechanisms in the tail of continuous variables, it does not address how causal structure behaves under *rare but non-extreme* events as rarity does not imply extremeness: events may be statistically rare while occurring at non-tail values (Ebtekar et al., 2025), such as values that are unexpectedly close to zero or values of highly unbalanced binaries. Consequently, approaches based on asymptotic extremes are not directly applicable in this setting. In contrast, we study causal contributions for rare events without relying on asymptotic limits. Our framework does not assume continuous variables and applies to variables defined over arbitrary spaces, such as discrete domains. In addition, related work on **mediation analysis** has primarily addressed decomposing causal effects into path-specific components in order to quantify how effects propagate along different causal pathways (Robins et al., 2022; Kawakami & Tian, 2025; Singal & Michailidis, 2024). While this literature enables attribution of portions of a causal effect to specific paths, it does not address a complementary question: which parts of a causal model provide a good causal explanation for a particular observed event. Therefore, our goal is not mediation analysis, but rather event-level interpretability of

causal structure for rare events. Another related line of work is **causal abstraction**, which provides a formal framework for relating causal models that represent the same system at different levels of granularity (Rubenstein et al., 2017; Beckers & Halpern, 2019; Beckers et al., 2020). However, existing approaches to causal abstraction operate at the level of the entire causal model, whereas our paper formalizes an event-level notion of causal pathway abstraction for the portion of a causal graph that explains a target rare event.

The paper is structured as follows. To quantify the extent to which a set of root-cause events explains a set of observed events, Section 2 discusses causal relations between binary variables. Section 3 defines *causal pathways* from root causes to a target event and quantifies to what extent the pathway explains the target. Section 4 describes how causal pathways of events, represented as binary variables, are obtained via abstraction of more detailed causal models. Section 5 demonstrates how our concepts can be applied to evaluate causal explanations.

## 2. Explaining Clusters of Events

Assume we are given $k$ events, formalized by binary variables $\mathbf{B} := \{B_1, \ldots, B_k\}$, where $B_i = 1$ denotes event $i$ happening. Further assume that the variables $\mathbf{B}$ are connected by a causal DAG $\mathcal{C}$ with joint probability distribution $P_{\mathbf{B}}$. For an arbitrary index set $S \subseteq K := \{1, \ldots, k\}$, let $\mathbf{B}_S$ denote the vector of variables $(B_i)_{i \in S}$, and let $\mathbf{B}_S = \mathbf{1}$ denote the joint event $(B_i = 1)_{i \in S}$. Then, assuming the Markov condition (Pearl, 2000), the probability of observing the event $P_{\mathbf{B}}(\mathbf{B} = \mathbf{1})$ factorizes as

$$P_{\mathbf{B}}(\mathbf{B} = \mathbf{1}) = \prod_{i=1}^{k} P_{\mathbf{B}}(B_i = 1 \mid \mathbf{B}_{\text{Pa}(i)} = \mathbf{1}), \quad (1)$$

where $\text{Pa}(i)$ denotes the indices of the parents of $B_i$ in $\mathcal{C}$. Now consider the scenario that an external event "corrupted" the data generating process in the sense that one or several of the causal mechanisms $P_{\mathbf{B}}(B_i \mid \mathbf{B}_{\text{Pa}(i)})$ are replaced with different mechanisms $\tilde{P}_{\mathbf{B}}(B_i \mid \mathbf{B}_{\text{Pa}(i)})$. Let $R \subseteq K$ denote the set of indices of the affected nodes. We define the variables $\mathbf{B}_R$ as the root-cause variables of the event $\mathbf{B} = \mathbf{1}$, and call the tuple $(\mathcal{C}, \mathbf{B}, \mathbf{B}_R, P_{\mathbf{B}})$ a *cluster of events* with root causes $R$. The "hard intervention" (Pearl, 2000) $do(\mathbf{B}_R = \mathbf{1})$ is a special case of the more general class of "soft interventions" (Eberhardt & Scheines, 2007), which yields

$$P_{\mathbf{B}}(\mathbf{B} = \mathbf{1} | do(\mathbf{B}_R = \mathbf{1})) = \prod_{i \notin R} P_{\mathbf{B}}(B_i = 1 | \mathbf{B}_{\text{Pa}(i)} = \mathbf{1}). \quad (2)$$

For the case of soft interventions, the right-hand side of (2) is only an upper bound on the likelihood of the cluster event

$\mathbf{B} = \mathbf{1}$, because the term $\prod_{i \in R} \tilde{P}_{\mathbf{B}}(B_i = 1 \mid \mathbf{B}_{\text{Pa}(i)} = \mathbf{1})$ is missing. Whenever the right-hand side of (2) is close to 1, the intervention $do(\mathbf{B}_R = \mathbf{1})$ is a *plausible explanation* for the cluster event. To formalize this idea, we first introduce a concept from Oesterle et al. (2025) with slightly different notation:

**Definition 2.1** (explanation score). Let $Y$ be a discrete variable and $\mathbf{X}$ a vector of variables influencing $Y$. For values $x$ and $y$, the event $\mathbf{X} = \mathbf{x}$ explains the event $Y = y$ with explanation score

$$\mathcal{E}(\mathbf{x} \to y) := 1 - \frac{\log P_{\mathbf{B}}(Y = y \mid do(\mathbf{X} = \mathbf{x}))}{\log P_{\mathbf{B}}(Y = y)}. \quad (3)$$

We call the quantity $1 - \mathcal{E}(\mathbf{x} \to y)$ the *explanation deficit*.

Following this terminology we define:

**Definition 2.2** (cluster explanation score). For any set $R \subseteq K$ of potential root causes, the event $\mathbf{B}_R = \mathbf{1}$ explains the cluster event $\mathbf{B} = \mathbf{1}$ with explanation score

$$\begin{aligned} \mathcal{E}_{R \to K} : \ &= \ 1 - \frac{\log P_{\mathbf{B}}(\mathbf{B} = \mathbf{1} \mid do(\mathbf{B}_R = \mathbf{1}))}{\log P_{\mathbf{B}}(\mathbf{B} = \mathbf{1})} \\ &= \ \frac{\sum_{i \in R} \log P_{\mathbf{B}}(B_i = 1 \mid \mathbf{B}_{\text{Pa}(i)} = \mathbf{1})}{\log P_{\mathbf{B}}(\mathbf{B} = \mathbf{1})}. \end{aligned}$$
$$(4)$$

Good candidates for root causes are therefore given by sets $R$ with high explanation score. Furthermore, while Oesterle et al. (2025) emphasize that explanation score is–for good reasons–not generally additive over disjoint unions of features, one can easily check that $\mathcal{E}_{R \to K}$ does have this property:

**Lemma 2.3** (additivity of contributions). *Let $R$ and $R'$ be disjoint subsets of $K$. Then*

$$\mathcal{E}_{R \cup R' \to K} = \mathcal{E}_{R \to K} + \mathcal{E}_{R' \to K}. \quad (5)$$

*Denoting $\mathcal{E}_{i \to K}$ instead of $\mathcal{E}_{\{i\} \to K}$, one can also verify that*

$$\sum_{i \in K} \mathcal{E}_{i \to K} = 1. \quad (6)$$

We call $\mathcal{E}_{i \to K}$ the *contribution* of mechanism $P_{\mathbf{B}}(B_i \mid \mathbf{B}_{\text{Pa}(i)})$ to the cluster event. By the additivity stated in Lemma 2.3, for every $R \subseteq K \setminus \{i\}$, we also have $\mathcal{E}_{i \to K} = \mathcal{E}_{R \cup \{i\} \to K} - \mathcal{E}_{R \to K}$. Hence, the contribution of each mechanism $i$ to the cluster explanation does not depend on which other root causes are considered simultaneously. Consequently, there is no need to average over contexts of variables to condition on as in Shapley-value "fair attribution" (Shapley, 1953), in contrast to causal and statistical attribution settings where marginal contributions may depend on the chosen context (Lundberg & Lee, 2017; Frye

et al., 2020; Wang et al., 2021; Janzing et al., 2020). In addition, note that for any discrete distribution $Q$, $-\log Q(x)$ can be written as the KL-divergence $D_{KL}(\delta_x \| Q)$, where $\delta_x$ is the point mass on $x$, which enables rewriting $\mathcal{E}_{R \to K}$ in terms of KL-divergences[1]. Using this observation, Appendix A.1 shows that $\mathcal{E}_{R \to K}$ can be seen as a degenerate instance of distribution change attribution in the sense of Budhathoki et al. (2021).

Although we use the term "event" throughout the paper, which may suggest occurrences at specific *time instants*, our framework more generally applies to explaining why a specific *statistical unit* behaves differently from a reference population, for instance an individual patient with a monogenic disorder may be viewed as an outlier relative to healthy individuals (Li et al., 2025).

## 3. Causal Pathways

We now focus on explaining a certain target event $B_t = 1$ within a cluster, and therefore focus on the subset of nodes that are crucial to understand how the target event has been triggered by the root causes *via a causal pathway*:

**Definition 3.1** (pathway explanation of events). A causal pathway is a tuple $(\mathcal{C}, \mathcal{P}, \mathbf{B}, B_t, \mathbf{B}_R, P_{\mathbf{B}})$ where
(i) $\mathcal{C}$ is a DAG describing the causal relations between the binary variables $\mathbf{B} := \{B_1, \ldots, B_k\}$, and the probability distribution $P_{\mathbf{B}}$ is therefore Markov relative to $\mathcal{C}$,
(iii) $\mathbf{B}_R \subseteq \mathbf{B}$ is the set of root causes, where $\mathbf{B}_R$ can also be empty or only contain $B_t$,
(iv) $\mathcal{P}$ is a subgraph of $\mathcal{C}$, the *pathway* which coincides with $\mathcal{C}$ except possibly for some edges into $R$,
(ii) $B_t \in \mathbf{B}$ is the unique sink node of $\mathcal{P}$ formalizing the event to be explained.

To understand the distinction between $\mathcal{P}$ and $\mathcal{C}$, observe that knowledge of $\mathcal{P}$ alone suffices to compute $P_{\mathbf{B}}(\mathbf{B}|do(\mathbf{B}_R = \mathbf{b}_R))$ for all $\mathbf{b}_R$, since the do-intervention removes all incoming edges to $R$. Thus, $\mathcal{P}$ is sufficient for explaining why the identified root causes render the target likely. However, computing the outcome when some root causes remain unintervened upon—as distinct from setting them to zero—requires knowledge of the incoming edges to $R$. In this regard, $\mathcal{C}$ enables assessment of each individual root cause's relevance and enhances the transparency of the explanation by allowing evaluation of whether the selected root causes are well-justified. In essence, $\mathcal{C}$ provides a meta-level component to the explanation by elucidating the causal pathway itself. Nevertheless, one can readily verify that the specific choice of $\mathcal{C}$ does not affect the scores $\mathcal{E}_{R \to t}^K$ and

---

[1]Note that (Oesterle et al., 2025) already introduced the explanation score as a KL divergence, as well as generalizations in terms of other distances between distributions.

$\mathcal{E}_{R \to t}$, which are defined later in this section.

Note, however, that $\mathcal{P}$ may also contain edges into nodes in $R$. This allows us to describe, for instance, scenarios like *chains of root causes*, in which each one "contributes" to the target because it does *not* propagate the perturbation as expected and adds an additional perturbation (or additional "unexpectedness") instead, see Example 3.6.

In the rest of this paper, we will refer to $\mathcal{P}$ as the causal pathway, when the other "ingredients" are clear from the context. To quantify to what extent the root causes explain the target event we again use the concept of explanation score:

**Definition 3.2** (target explanation score).

$$\mathcal{E}_{S \to t} := 1 - \frac{\log P_{\mathbf{B}}(B_t = 1 \mid do(\mathbf{B}_S = \mathbf{1}))}{\log P_{\mathbf{B}}(B_t = 1)}, \quad (7)$$

where $S$ can be any subset of nodes.

While Definition 3.2 can be used to measure relevance to the target for any set of nodes, we will mostly use $\mathcal{E}_{R \to t}$ to measure the relevance of the root causes. However, $\mathcal{E}_{R \to t}$ does not capture the extent to which the remaining events in $K \setminus (R \cup \{t\})$ are crucial for explaining the target event–in contrast with common intuitive notions of a causal pathway, which entail some relevance of all events mentioned in a verbal explanation. This motivates the following concept, which is a modification of the explanation score:

**Definition 3.3** (pathway explanation score). The pathway is said to explain the target with pathway explanation score

$$
\begin{aligned}
\mathcal{E}_{R \to t}^K & := 1 - \frac{\log P_{\mathbf{B}}(\mathbf{B} = \mathbf{1} \mid do(\mathbf{B}_R = \mathbf{1}))}{\log P_{\mathbf{B}}(B_t = 1)} \\
& = 1 - \frac{\log P_{\mathbf{B}}(\mathbf{B}_{K \setminus R} = \mathbf{1} \mid do(\mathbf{B}_R = \mathbf{1}))}{\log P_{\mathbf{B}}(B_t = 1)}.
\end{aligned}
\tag{8}
$$

Definition 3.3 formalizes the idea that making the root cause events happen renders all the events "relatively likely", where likelihood is assessed relative to the "unlikeliness" of the target. Note that $\mathcal{E}_{R \to t}^K$ is negative if the likelihood of $\mathbf{B} = \mathbf{1}$, given the root causes $\mathbf{B}_R = \mathbf{1}$, is smaller than the a priori likelihood of $B_t = 1$–in other words the explanation "explains" the target event with an even more unlikely cluster of events. The following result shows the difference between Definitions 3.3 and 3.2:

**Lemma 3.4** (pathway score versus target score). *If $S$ denotes the complement of $R \cup \{t\}$, we have*

$$
\begin{aligned}
& \mathcal{E}_{R \to t} - \mathcal{E}_{R \to t}^K \\
& = \frac{\log P_{\mathbf{B}}(\mathbf{B}_S = \mathbf{1} \mid B_t = 1, do(\mathbf{B}_R = \mathbf{1}))}{\log P_{\mathbf{B}}(B_t = 1)} \geq 0. \quad (9)
\end{aligned}
$$

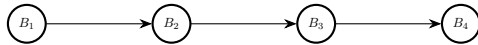

*Figure 1.* Example showing that whether a node should be considered a root cause does not depend on its position in the DAG: to show this, the text describes parameters for which $B_1$ and $B_3$ should be considered root causes, while $B_4$ is only a root cause if we target a higher pathway explanation score.

It is important to note that $\mathbf{B}_S$ need not necessarily be mediators that propagate the information from $\mathbf{B}_R$ to $B_t$. We will later also have cases where they describe "context" events that control the propagation of information from $\mathbf{B}_R$ to $B_t$ as in Example 4.8. (9) measures to what extent configurations other than $\mathbf{B}_S = \mathbf{1}$ are relevant when the intervention $do(\mathbf{B}_R = \mathbf{1})$ changes the probability of $B_t = 1$. For a "good" pathway, we expect (9) to be small, because all its events should occur with high probability when the root cause variables are set to $1$ and the target event is observed to happen. Lemma 3.4 entails in particular that $\mathcal{E}_{R \to t}^K > 0$ implies $\mathcal{E}_{R \to t} > 0$, and thus guarantees that root causes with positive pathway explanation score are indeed relevant for the *target*, and not only for the other nodes in the pathway.

Pathway explanation score and cluster explanation score are related as follows:

**Lemma 3.5** (pathway score versus cluster score)**.**

$$1 - \mathcal{E}_{R \to t}^K = (1 - \mathcal{E}_{R \to K}) \cdot \frac{\log P_{\mathbf{B}}(\mathbf{B} = \mathbf{1})}{\log P_{\mathbf{B}}(B_t = 1)}. \qquad (10)$$

Lemma 3.5 shows an affine relation between the two types of scores, which implies the modified additivity

$$\mathcal{E}_{R \to t}^K - \mathcal{E}_{\emptyset \to t}^K = \sum_{i \in R}(\mathcal{E}_{\{i\} \to t}^K - \mathcal{E}_{\emptyset \to t}^K). \qquad (11)$$

Lemma 3.5 also shows that the pathway explanation score defines a stricter assessment of an explanation compared to the cluster explanation score: an explanation is considered good only if $P_{\mathbf{B}}(\mathbf{B} = \mathbf{1} \mid do(\mathbf{B}_R = \mathbf{1}))$ is much closer to $1$ than the baseline probability $P_{\mathbf{B}}(B_t = 1)$, while the cluster explanation score is close to $1$ if $P_{\mathbf{B}}(\mathbf{B} = \mathbf{1} \mid do(\mathbf{B}_R = \mathbf{1}))$ is much closer to $1$ than the typically smaller baseline probability $P_{\mathbf{B}}(\mathbf{B} = \mathbf{1})$.

Example A.2 further illustrates the distinction between pathway and cluster explanation scores.

**Example 3.6** (chain of events)**.** Let $\mathcal{P}$ be the DAG in Figure 1, where each of the variables $B_2, B_3, B_4$ can attain value $1$ only if its parent attains value $1$. Here and later, for a binary variable $B_j$, we write $b_j^a$ as shorthand for the event $B_j = a$, where $a \in \{0, 1\}$. For the probability mass function $p_{\mathbf{B}}$, we assume $p_{\mathbf{B}}(b_1^1) = 10^{-3}$, $p_{\mathbf{B}}(b_2^1 \mid b_1^1) = 1$, $p_{\mathbf{B}}(b_3^1 \mid b_2^1) = 10^{-3}$, $p_{\mathbf{B}}(b_4^1 \mid b_3^1) = 10^{-2}$. Thus, the target event $b_4^1$ occurs only if all other events $b_1^1, b_2^1, b_3^1$ occur,

which happens with probability $1/10^8$ only. In this case we would say that the nodes $B_1$ and $B_3$ behaved particularly unexpectedly, since they showed a behavior that they only show in 1 out of $1{,}000$ cases. The node $B_2$ behaved as usual, since $b_2^1$ occurs with probability 1 conditional on $b_1^1$. The target event $b_4^1$ is rare marginally, but the mechanism producing it is only moderately surprising: conditional on $b_3^1$, the event $b_4^1$ occurs with probability $10^{-2}$, which is not that rare compared to how rare the target event is. Thus, we get pathway explanation score $\mathcal{E}_{R \to t}^K = 1 - \log 10^2 / \log 10^8 = 3/4$, if we take only $B_1$, $B_3$ as root causes, but pathway explanation score $\mathcal{E}_{R \to t}^K = 1$, if $B_4$ is also added to the root causes.

In Appendix A.3 we show that even when each event strongly explains the next one in a causal chain, this does not guarantee that the initial event has a high pathway explanation score. Therefore, strong local causal explanations do not necessarily imply a strong global causal explanation. The following result describes a necessary condition for the pathway explanation score to be close to 1:

**Lemma 3.7** (no large log-likelihood gaps)**.** *For any node $i$ we define the log-likelihood gap*

$$\Delta_i := [\log P_{\mathbf{B}}(B_{\mathrm{Pa}(i)} = \mathbf{1}) - \log P_{\mathbf{B}}(B_i = 1)]_+,$$

*where* $[a]_+ := (a + |a|)/2$. *Then we have*

$$\mathcal{E}_{R \to t}^K \leq 1 - \frac{\sum_{i \notin R} \Delta_i}{- \log P_{\mathbf{B}}(B_t = 1)}. \qquad (12)$$

The proof follows easily from $P_{\mathbf{B}}(B_i = 1 \mid \mathbf{B}_{\mathrm{Pa}(i)} = \mathbf{1}) \leq P_{\mathbf{B}}(B_i = 1)/P_{\mathbf{B}}(\mathbf{B}_{\mathrm{Pa}(i)} = \mathbf{1})$, or as a special case of Lemma 3.3 in Orchard et al. (2025). Lemma 3.7 states that good explanations need to avoid events as mediators that are significantly less likely than their parent event. As a simple application of Lemma 3.7, we also get a low explanation score when a mediator event is less rare than the target since we get a significant gap for the target itself, as in Example A.3.

**Relation to probability of necessity and sufficiency.** At first glance, the notion of explaining pathway seems to miss the aspect of *necessity*: while it formalizes the condition that the target event and the mediators between root causes and target are likely to happen given the root causes happened, it does not explicitly state that the target event is unlikely to happen in the absence of the root causes. Likewise, one may be surprised that explanation score of Oesterle et al. (2025) in general fails to formalize necessity. However, there is a sense in which necessity is implicit because the target event is rare: the intervention $do(\mathbf{B}_R = \mathbf{1})$ is, with high probability necessary for $B_t = 1$ because without it the latter is unlikely by assumption. This is formally shown in Appendix B, based on the notion *probability of necessity*

(Tian & Pearl, 2000) (where we also argue why probability of *sufficiency* is of minor relevance in our context). This result, however, does not imply that it is necessary to set *all* root causes to 1. The choice of a sufficiently small set of root causes will be driven by trading off simplicity of the explanation (in the sense of a small number of root causes) with pathway explanation score.

In Appendix C we explain why our causal explanations refer to interventional probabilities only, rather than causal counterfactuals in the sense of rung 3 in the ladder of causation (Pearl & Mackenzie, 2018).

**Measuring quality of explanations.** We now state formal criteria that should be satisfied by a causal pathway to be a good explanation:

*1. Causal relevance of all events for the target:* Given some cluster $(\mathcal{C}, \mathbf{B}, \mathbf{B}_R, P_{\mathbf{B}})$, every event $i \in K$ should satisfy $\mathcal{E}_{\{i\} \cup S \to t} - \mathcal{E}_{S \to t} > 0$ for some $S \subset K$. Depending on one's preferences regarding the trade-off between simplicity and detail, one may set a threshold for the desired increase in explanation score. Note that we do *not* consider *pathway* explanation scores $\mathcal{E}_{R \to t}^K$ for this purpose because we want to assess relevance for the target and not for other events in the pathway.

*2. Sparsity of root causes:* Following Occam's Razor we a priori prefer explanations with a small number of root causes. Whenever the target event is the result of the respective data point *being drawn from a different distribution* this inductive bias is also aligned with the "sparse mechanism shift hypothesis" in (Schölkopf et al., 2021). However, rare events may also originate from *selecting* the statistical unit / time instant for which this rare event happened. Later we will have target events that come from thresholding continuous variables and the statistical unit we discuss is selected because it is the most extreme value within a population. Rare events that result from this selection bias are not necessarily points that are "out of distribution". In this case, we may also expect multiple root causes. For instance, if $X_1, \ldots, X_d$ are independent identically distributed Gaussians, extreme values of $\sum_{i=1}^{d} X_i$ likely originate from extreme values of multiple $X_i$, while $X_i$ with subexponential $X_i$ render single root causes more likely, see Appendix E.

*3. Selecting the right root causes:* The pathway explanation score $\mathcal{E}_{R \to t}^K$ should be reasonably close to 1, without unnecessarily extending $R$. For any desired size $|R|$, (11) implies that the best $R$ can be obtained by a greedy approach in a scenario where the set $\mathbf{B}$ is fixed: starting from $R = \emptyset$, add the event $i$ to $R$ that maximizes the score $\mathcal{E}_{\{i\} \cup R \to t}^K$. After choosing $R$, we can reduce the graph $\mathcal{C}$ to the relevant part, the pathway $\mathcal{P}$, by dropping arrows into $R$ (although we may not drop all of them, as explained earlier).

## 4. Pathway Abstraction

We now introduce causal pathways that constitute approximate abstractions of more fine-grained causal models. The abstraction proceeds by reducing the set of nodes and mapping the remaining variables $X_i$ into a binary representation. In addition, edges may be added or removed to provide the best possible approximation of the causal relationships among the resulting binary variables.[2]

### 4.1. Binary Representation of Variables

Let us first focus on generating binaries from variables with arbitrary ranges. Consider, for instance, a cause-effect pair $X, Y$ coupled by the linear structural equation $Y = \alpha X + N$, where $N$ is an independent noise term and all three variables are real-valued. If we induce a large perturbation $x$ and accordingly obtain an unusually large $y$, we would certainly accept the verbal explanation "$y$ is large because $x$ has been large". To get formal criteria for whether this explanation is plausible we introduce the events $B_1$ and $B_2$ corresponding to the events[3] $X \geq x$ and $Y \geq y$, and consider only accepting the explanation if $P(Y \geq y | X \geq x)$ is "sufficiently large" in the sense specified below. In cases where $Y$ depends on $X$ in a non-monotonic way, for instance, when $Y = \sin(X) + N$, we would not accept explanations like "$y$ is large because $x$ is large" and instead suggest the "featurized" version "$y$ is large because $\sin(x)$ is large" to get a "feature-monotonic" causal relation, formally introduced here[4]:

**Definition 4.1** (feature monotonicity). Consider a causal DAG $\mathcal{G}$ with nodes $\mathbf{X} = \{X_1, \ldots, X_n\}$ and probability distribution $P_{\mathbf{X}}$, such that each $X_i$ attains values in $\mathcal{X}_i$. For any $I \subseteq N := \{1, \ldots, n\}$, define feature functions $\tau_I(\mathbf{X}_I) := (\tau_i(X_i))_{i \in I}$, where $\tau_i : \mathcal{X}_i \to \mathbb{R}$ for all $i \in I$. Further, write $\tau_I(\mathbf{x}'_I) \geq \tau_I(\mathbf{x}_I)$ iff $\tau_i(x'_i) \geq \tau_i(x_i)$ for all $i \in I$. Then the Markov kernel $P_{\mathbf{X}}(X_j \mid \mathbf{X}_{\mathrm{Pa}(j)})$ satisfies feature monotonicity (with respect to $\tau_j, \tau_{\mathrm{Pa}(j)}$) if

$$P_{\mathbf{X}}\big(\tau_j(X_j) \geq t \mid \mathbf{x}'_{\mathrm{Pa}(j)}\big) \geq P_{\mathbf{X}}\big(\tau_j(X_j) \geq t \mid \mathbf{x}_{\mathrm{Pa}(j)}\big), \tag{13}$$

for all $t \in \mathbb{R}$, whenever $\tau_{\mathrm{Pa}(j)}(\mathbf{x}'_{\mathrm{Pa}(j)}) \geq \tau_{\mathrm{Pa}(j)}(\mathbf{x}_{\mathrm{Pa}(j)})$.

Here, featurizing variables has two goals. First, it defines *outlier scores* (Budhathoki et al., 2022b) via $S(x_j) :=$

---

[2]We require approximate equality of interventional probabilities, cf. related notions of approximate abstraction in the literature (Beckers et al., 2020). Here, however, "approximate" is defined by our logarithmic scaling, which is particularly tailored to rare events.

[3]For notational convenience, we avoid superscripts like in $B_1^x, B_2^y$, but keep the dependence on $x$ and $y$ in mind.

[4]cf. monotonicity condition in Lemma 3.5 in Orchard et al. (2025).

$-\log P_{\mathbf{X}}(\tau_j(X_j) \geq \tau_j(x_j))$, which measures the unexpectedness of events for arbitrary data modalities. Second, it enables causal statements that are more robust than statements about the impact of setting a continuous variable to a specific value, cf. Díaz et al. (2023). We then obtain:

**Lemma 4.2** (bounding likelihood for each Markov kernel). *Let* $\mathbf{x}_{\mathrm{Pa}(j)}$ *be arbitrary*, $x_j$ *drawn from* $P_{\mathbf{X}}(X_j \mid \mathbf{X}_{\mathrm{Pa}(j)} = \mathbf{x}_{\mathrm{Pa}(j)})$ *and assume that* $P_{\mathbf{X}}(X_j \mid \mathbf{X}_{\mathrm{Pa}(j)})$ *satisfies feature monotonicity. Then, for any* $\alpha \in [0,1]$, *the probability for obtaining an* $x_j$ *with* $P_{\mathbf{X}}\big(\tau_j(X_j) \geq \tau_j(x_j) \mid \tau_{\mathrm{Pa}(j)}(\mathbf{X}_{\mathrm{Pa}(j)}) \geq \tau_{\mathrm{Pa}(j)}(\mathbf{x}_{\mathrm{Pa}(j)})\big) \leq \alpha$, *is at most* $\alpha$.

Note that the non-trivial part of the statement comes from conditioning on the event $\tau_{\mathrm{Pa}(j)}(\mathbf{X}_{\mathrm{Pa}(j)}) \geq \tau_{\mathrm{Pa}(j)}(\mathbf{x}_{\mathrm{Pa}(j)})$ instead of $\mathbf{X}_{\mathrm{Pa}(j)} = \mathbf{x}_{\mathrm{Pa}(j)}$ used in the generating process for $x_j$. However, feature monotonicity entails

$$P_{\mathbf{X}}\big(\tau_j(X_j) \geq \tau_j(x_j) \mid \tau_{\mathrm{Pa}(j)}(\mathbf{X}_{\mathrm{Pa}(j)}) \geq \tau_{\mathrm{Pa}(j)}(\mathbf{x}_{\mathrm{Pa}(j)})\big)$$
$$\geq P_{\mathbf{X}}\big(\tau_j(X_j) \geq \tau_j(x_j) \mid \mathbf{X}_{\mathrm{Pa}(j)} = \mathbf{x}_{\mathrm{Pa}(j)}\big), \quad (14)$$

which then completes the proof. To get an intuition about the implications of Lemma 4.2, we consider a cause-effect pair $X \to Y$ with feature monotonic $P(Y \mid X)$ with respect to $\tau_X, \tau_Y$. For any $x$-value, a $y$-value drawn from $P(Y \mid X = x)$ satisfies $-\log P\big(\tau_Y(Y) \geq \tau_Y(y) \mid \tau_X(X) \geq \tau_X(x)\big) \geq c$ with probability at most $e^{-c}$. Accordingly, defining the binary events $\tau_X(X) \geq \tau_X(x)$ and $\tau_Y(Y) \geq \tau_Y(y)$ typically results in explanation scores close to 1 when the generated value $y$ is a "strong outlier", that is, $P(\tau_Y(Y) \geq \tau_Y(y))$ is small.

The following result generalizes Lemma 4.2 to general DAGs:

**Theorem 4.3** (bounding multivariate likelihoods). *For any* $\mathbf{x}_R$, *let* $\bar{R} := N \setminus R$, *and let* $\mathbf{x}_{\bar{R}}$ *be generated from* $P_{\mathbf{X}}(\mathbf{X}_{\bar{R}} \mid do(\mathbf{x}_R))$. *Assume that for all* $j \in N$, *the mechanisms* $P_{\mathbf{X}}(X_j \mid \mathbf{X}_{\mathrm{Pa}(j)})$ *are feature monotonic w.r.t. the feature functions* $\tau_1, \ldots, \tau_n$ *and let* $B_j$ *denote the event* $\tau_j(X_j) \geq \tau_j(x_j)$. *Then the probability of obtaining an* $\mathbf{x}$ *with*

$$L := -\sum_{j \notin R} \log P_{\mathbf{X}}(B_j = 1 \mid \mathbf{B}_{\mathrm{Pa}(j)} = \mathbf{1}) \geq c, \quad (15)$$

*is at most*

$$p = \sum_{i=0}^{n-|R|-1} \frac{c^i}{i!} e^{-c}. \quad (16)$$

The proof is provided in Appendix F. To understand the significance of Theorem 4.3, consider the joint distribution[5] $P_{\mathbf{B}}(\mathbf{B}) := \prod_{j \in N} P_{\mathbf{X}}(B_j \mid \mathbf{B}_{\mathrm{Pa}(j)})$. Note that $P_{\mathbf{B}}(\mathbf{B})$ is, by construction, Markov relative to $\mathcal{G}$, whereas the true distribution $P_{\mathbf{X}}(\mathbf{B})$ need not be; even for $X_1 \to X_2 \to$

---

[5]For any $S$, we denote by $P_{\mathbf{X}}(\mathbf{B}_S)$ the pushforward measure under our binarization map *for fixed* $\mathbf{x}_S$.

$X_3$, binarization can destroy the conditional independence $X_1 \perp\!\!\!\perp X_3 \mid X_2$. Let us, nevertheless, consider a scenario where $P_{\mathbf{B}}(\mathbf{B})$ is a reasonable approximation for $P_{\mathbf{X}}(\mathbf{B})$, because we can easily compute interventional probabilities for the former. We then observe that $L$ is the negative log-likelihood of $P_{\mathbf{B}}(\mathbf{B} = \mathbf{1} \mid do(\mathbf{B}_R = \mathbf{1}))$, and thus it follows from Theorem 4.3 that the pathway explanation score is close to 1 with high probability, whenever $\mathbf{x}_R$ induces a value $x_t$ of the target node such that $P_{\mathbf{B}}(B_t = 1)$ is small. Although feature monotonicity may not hold in general, we assume that *good* explanations should use features that do not deviate too much from monotonicity and use the p-value in (16) as guidance on which deviations from explanation score 1 we tolerate for the given number $n$ of nodes, regardless of whether feature monotonicity holds exactly for a given case.

### 4.2. Causal Pathways as Approximate Abstractions

To enable simple causal explanations, we introduce pathways (potentially with fewer links and variables than the original graph) with distributions that do not perfectly align with the true one, but do not deviate too much in terms of interventional and observational probabilities. This raises the issue that the interventions $do(B_i = b_i)$, with $b_i = 0, 1$, are only well-defined in the model $P_{\mathbf{B}}$, but are ill-defined in $P_{\mathbf{X}}$ since there are generally many distinct interventions $do(X_j = x_j)$ that correspond to the same value of $B_j$. This aligns with the known ill-definedness of interventions on coarse-grained variables (Spirtes & Scheines, 2004). Consider, for instance, the event $X_j \geq x_j$ for some unusually large $x_j$. Then $do(X_j = x_j - \epsilon)$, which sets $B_j$ to 0, will typically result in distributional changes that are close to $do(X_j = x_j)$, which sets $B_j$ to 1, and thus far from the effect of "normal" $x_j$. We solve this problem by interpreting $do(\mathbf{B}_j = \mathbf{b}_j)$ in the original model as a randomized intervention that draws $X_j$ from the distribution $P_{\mathbf{X}}(X_j \mid B_j = b_j)$, following the idea of "natural micro-realization" given by Definition 8 in Zhu et al. (2024). For multi-node interventions we draw independently from the product distribution $\prod_{i \in S} P_{\mathbf{X}}(X_i \mid B_i = b_i)$ and thus define:

**Definition 4.4** (natural micro-realization). The natural micro-realization of $do(\mathbf{B}_S = \mathbf{b}_S)$ in $\mathcal{G}$ is given by

$$P_{\mathbf{X}}(\mathbf{X} \mid do(\mathbf{B}_S = \mathbf{b}_S))$$
$$:= \mathbb{E}_{\mathbf{x}_S \sim \prod_{i \in S} P_{\mathbf{X}}(X_i \mid B_i = b_i)}[P_{\mathbf{X}}(\mathbf{X} \mid do(\mathbf{X}_S = \mathbf{x}_S))]. \quad (17)$$

For an unconfounded cause-effect pair $X \to Y$, for instance, this definition ensures that $P(Y \mid do(X \geq x)) = P(Y \mid X \geq x)$. We now introduce pathway abstraction formally:

**Definition 4.5** (pathway abstraction). A pathway abstraction of a causal DAG $\mathcal{G}$, with nodes $\mathbf{X} = \{X_1, \ldots, X_n\}$

and probability distribution $P_{\mathbf{X}}$, is a pathway explanation $(\mathcal{C}, \mathcal{P}, \mathbf{B} = \{B_1, \ldots, B_k\}, B_t, \mathbf{B}_R, P_{\mathbf{B}})$, where $P_{\mathbf{B}}$ is Markov relative to $\mathcal{C}$. For each $j \in K$, there exists a subset $I_j \subseteq N$ and feature functions $\tau_{I_j}(\mathbf{X}_{I_j}) = (\tau_{i_j}(X_{i_j}))_{i_j \in I_j}$ with $\tau_{i_j} : \mathcal{X}_{i_j} \to \mathbb{R}$ for all $i_j \in I_j$, such that for an observed value $\mathbf{x}_{I_j}$, the binary variable $B_j$ is defined by $B_j := \chi_{[\tau_{I_j}(\mathbf{x}_{I_j}), \infty)}(\tau_{I_j}(\mathbf{X}_{I_j}))$. The explanation score of the pathway abstraction is defined as the explanation score of $\mathcal{P}$, and the accuracy of the pathway abstraction is defined by

$$r := 1 - \max_{S \subseteq K} \max_{\mathbf{b}_S} \left\{ \frac{D_{KL}[P_{\mathbf{X}}(\mathbf{B} \mid do(\mathbf{B}_S = \mathbf{b}_S)) \| P_{\mathbf{B}}(\mathbf{B} \mid do(\mathbf{B}_S = \mathbf{b}_S))]}{-\log P_{\mathbf{X}}(B_t = 1)} \right\}, \tag{18}$$

where $S = \emptyset$ is also included to capture observational probabilities too.

Note that the pathway explanation score can also be rephrased in terms of KL-divergences as

$$\mathcal{E}_{R \to t}^K = 1 - \frac{D_{KL}(\delta_{\mathbf{1}}(\mathbf{B}) \| P_{\mathbf{B}}(\mathbf{B} \mid do(\mathbf{B}_R = \mathbf{1})))}{-\log P_{\mathbf{B}}(B_t = 1)}, \tag{19}$$

where $\delta_{\mathbf{1}}$ denotes the point mass on $\mathbf{B} = \mathbf{1}$ (see remark after Definition 2.2). Hence, the explanation score measures the distance of $\delta_{\mathbf{1}}$ from the interventional distribution in $P_{\mathbf{B}}$ with respect to $\mathcal{P}$, where all root causes are set to 1, and accuracy measures the distance between the distribution of the abstraction and the true distribution projected to the binaries. In this sense, both quantities measure the quality of the explanation in terms of KL-divergences. Note that minimizing the KL-divergence (18) is equivalent to maximizing the expected log-likelihood $\mathbb{E}_{P_{\mathbf{X}}(\mathbf{B} \mid do(\mathbf{b}_S))}[\log P_{\mathbf{B}}(\mathbf{B} \mid do(\mathbf{b}_S))]$. Therefore trading off explanation score with accuracy amounts to trading off two likelihood terms, namely the one of the event $\mathbf{B} = \mathbf{1}$ under the model $P_{\mathbf{B}}(\mathbf{B} \mid do(\mathbf{B}_R = \mathbf{1}))$ and the likelihood of observations from $P_{\mathbf{X}}(\mathbf{B} \mid do(\mathbf{b}_S))$ under the models $P_{\mathbf{B}}(\mathbf{B} \mid do(\mathbf{b}_S))$.

### 4.3. Examples of Pathway Abstractions

We first start with a simple example where the abstraction consists only in binarization of two continuous variables:

**Example 4.6** (cause-effect pair). Assume $X \to Y$ has the linear structural model $Y = \rho X + N$, where $X, Y$ are standard Gaussian variables with joint probability distribution $p$, and $N$ is a Gaussian with variance $\sigma_N^2 := 1 - \rho^2$. Let $(x, y)$ be an arbitrary pair. Defining the feature functions $\tau_X(x) := x$ and $\tau_Y(y) := y$, we obtain binary variables $B_1$ and $B_2$ for the events $X \geq x$ and $Y \geq y$, respectively. For the pathway abstraction we set $P_{\mathbf{B}}(\mathbf{B}) := P(\mathbf{B})$,

where cluster $\mathcal{C}$ and pathway $\mathcal{P}$ are both defined by the DAG $B_1 \to B_2$, with root cause $B_1$ and target $B_2$. Due to Theorem 3.1 in (Zhu et al., 2024), the natural micro-realization entails $p_{\mathbf{B}}(b_2 \mid do(b_1)) = p(b_2 \mid do(b_1))$, hence, the pathway abstraction has accuracy 1. The explanation score reads

$$\mathcal{E}_{1 \to 2}^{\{1,2\}} = 1 - \frac{\log p(b_2^1 \mid b_1^1)}{\log p(b_2^1)} = \frac{\log \frac{P(X \geq x, Y \geq y)}{P(X \geq x)P(Y \geq y)}}{-\log P(Y \geq y)}.$$

Figure 2 shows isolines of the (pathway) explanation score for different pairs $(x, y)$ for $\rho = 0.5$. One can see that for outliers $x$ above 3, every $y$ below $\rho \cdot x$ yields $q > 0.8$. We propose to verbalize this by 'the event $X \geq x$ explains $Y \geq y$ up to at least 80%'. In the region with explanation score far below 50% we would reject the statement "$Y \geq y$ has been caused by $X \geq x$" and also the informal statement "$y$ is large because $x$ is large".

Example 4.6 shows that the pathway explanation score conceptualizes a certain kind of *robust* causal statements: It does *not* measure whether the specific value $x$ increases the probability of the event $Y \geq y$. Instead, it measures whether 'typical values' of $X$ from the interval $[x, \infty)$ have this effect, which is a more valuable insight since it is more general and falsifiable. The following example shows why *root nodes* in a pathway abstraction are not necessarily *root causes*, but potentially non-rare events that are nevertheless crucial as context.

**Example 4.7** (non-rare event as confounder). Let $B_1, B_2, B_3$ be binary variables with the causal structure in figure 3, middle. Let $B_1$ be an unbiased coin flip, and $P(B_2 = 1 \mid B_1 = 0) = 1$ and $P(B_2 = 1 \mid B_1 = 1) = \delta$ for some small $\delta > 0$. Moreover, $B_3 = B_1 \wedge B_2$. The target event $B_3 = 1$ is rare as it occurs with probability $\delta/2$ only. Although $B_2 = 1$ is not rare, the intervention $do(B_2 = 1)$ has a strong effect on the target and makes it happen with probability $1/2$. Here, the context $B_1 = 1$ is crucial to understand the big impact of the intervention on the target because $B_2 = 1$ occurring *together* with $B_1 = 1$ causes the unexpected behavior $B_3 = 1$. Hence we choose the graph in Figure 3, right, as pathway $\mathcal{P}$, and set $P_{\mathbf{B}}(\mathbf{B}) = P(\mathbf{B})$ but keep the full DAG in Figure 3, middle, as $\mathcal{C}$, since $P_{\mathbf{B}}$ is not Markov to $\mathcal{P}$. Here the pathway abstraction map is trivial and we thus obtain accuracy 1 and pathway explanation score $\mathcal{E}_{2 \to 3}^{\{1,2,3\}} = 1 - \frac{\log(1/2)}{\log(\delta/2)}$, which is close to 1 for small $\delta$. In contrast, the simple pathway $B_2 \to B_3$ would fail to explain why the frequent event $B_2 = 1$ triggered the rare event $B_3 = 1$, due to the missing context $B_1 = 1$. If we set $P_{\mathbf{B}}(B_2, B_3) = P(B_2, B_3)$ for the bivariate abstraction, then $P_{\mathbf{B}}(B_3 = 1 \mid do(B_2 = 1)) = P(B_3 \mid B_2 = 1) = \frac{\delta}{1+\delta}$, while $P_{\mathbf{B}}(B_3 = 1 \mid do(B_2 = 1)) = P(B_1) = \frac{1}{2}$. Thus

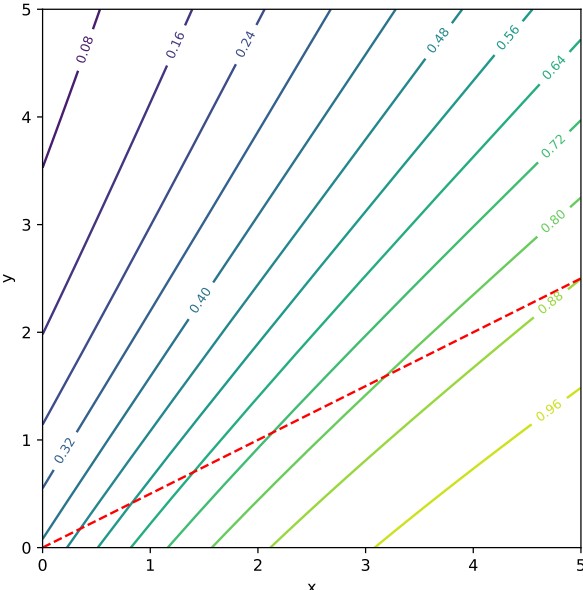

*Figure 2.* (Pathway) explanation scores $\mathcal{E}_{1\to2} = \mathcal{E}_{1\to2}^{\{1,2\}}$ for the binary cause effect pair obtained from thresholding a linear causal relation of Gaussians. For $y \approx \rho x$ (points near the red line) the explanation score reaches $80\%$ whenever $x$ is an outlier of strength 3 or more.

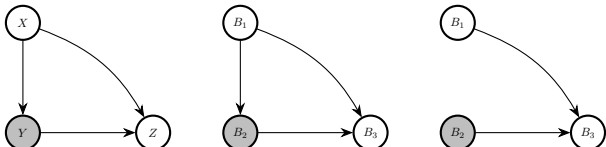

*Figure 3.* Left: DAG with three real-valued variables, where we perturb $Y$. Middle: the same DAG with binary variables $B_1, B_2, B_3$. To explain the impact of the intervention $do(B_2 = 1)$ (root cause node shown in gray) on the target $B_3$, we need a pathway $\mathcal{P}$ that also contains the confounder $B_1$ as context (right). In Example 4.8 where $B_1, B_2, B_3$ are binarizations of $X, Y, Z$ they stand for the events $|X| \leq |x|, Y \geq y, Z \geq z$.

the bivariate abstraction has accuracy

$$r = 1 - \frac{D_{KL}\left[\mathrm{Bern}\left(\frac{1}{2}\right) \,\middle\|\, \mathrm{Bern}\left(\frac{\delta}{1+\delta}\right)\right]}{-\log P(B_3 = 1)}$$

$$= 1 - \frac{\log \frac{1+\delta}{2\sqrt{\delta}}}{-\log(\delta/2)},$$

which converges to $1/2$ as $\delta \to 0$.

The following example shows a similar case obtained by binarization of continuous variables. We consider the causal DAG in Figure 3, left, where an extreme value $y$ is induced at $Y$. For understanding the impact of this perturbation on $Z$, it is important to know that $X$ attained normal values to exclude strong influence of the confounder in the perturbed regime. Hence, $X$ being normal is required as

*context* information. The example also shows that binarization can result in accuracy smaller than 1 because the natural micro-realization of the intervention does not capture the dependencies of different parents entirely. However, for our purpose of causal explainability, we do not need accuracy 1 in order to get simple and verbalizable explanations.

**Example 4.8** (confounder as context)**.** For the DAG in figure 3, left, assume linear equations

$$\begin{aligned} Y &= \alpha X + N_Y, \\ Z &= \beta X + \gamma Y + N_Z, \end{aligned} \qquad (20)$$

where $N_Y, N_Z$ are Gaussians, and $X, Y, Z$ are standard Gaussians with joint probability distribution $p$. We set $\alpha = -0.9$, $\beta = 0.9$, and $\gamma = 0.9$. Thus, $X$ negatively affects $Y$, while both $X$ and $Y$ positively affect $Z$, creating negative confounding between $Y$ and $Z$. Since we want to verbalize that large $y$ induced large $z$, we define feature functions $\tau_2(y) := y$ and $\tau_3(z) := z$ to capture large tail events and $\tau_1(x) := -|x|$ to describe that $x$ is "normal". The event $|X| \leq x$ formalizes the context in which the positive causal effect $Y \to Z$ is not offset by the confounding path through $X$. We will use the pathway $B_2 \to B_3 \leftarrow B_1$, where $B_2$ and $B_3$ are the root cause and target, respectively, while $\mathcal{C}$ contains $B_1 \to B_2$ in addition. Since every distribution is Markov to the complete DAG $\mathcal{C}$, we can set $P_\mathbf{B}(\mathbf{B}) := P(\mathbf{B})$, with $P$ shorthand for $P_{X,Y,Z}$. The pathway explanation score reads:

$$\begin{aligned} &\mathcal{E}_{2\to3}^{\{1,2,3\}} = \\ &1 - \frac{\log\left(P(Z \geq z \mid |X| \leq x, Y \geq y)P(|X| \leq x)\right)}{\log P(Z \geq z)}. \end{aligned}$$

We also consider the bivariate abstraction $B_2 \to B_3$, which omits the context variable $B_1$ and both the pathway and the cluster are given by the DAG $B_2 \to B_3$, and we set $P_\mathbf{B}(B_2, B_3) := P(B_2, B_3)$. Figure 4 in Appendix G.1 compares the the explanation scores and abstraction accuracies of the bivariate and trivariate abstractions for $x = 1$ and different values of $y, z$. Only in the regime where the explanation score is not *too far from* 1 would we accept the verbal explanation that "$z$ is as large because $y$ is large, in the context that $x$ is normal". The figure shows that the trivariate abstraction has higher accuracy and explanation score in the outlier region because the "context variable" $X$ (although being in a normal state) is necessary to understand the propagation of the perturbation of $Y$ to $Z$. Therefore, the pathway $B_2 \to B_3$ has lower accuracy because it does not include the context variable needed to approximate the relevant interventional distributions.

In Appendix G.2 we consider additional pathway abstractions for the three-variable binary causal DAG and give quantitative conditions under which the direct link or the

confounder can be removed while maintaining high abstraction accuracy.

## 5. Evaluating Causal Explanations

The assumption that all mechanisms of non-root causes operated as expected for the statistical observation to be explained entails testable implications. Phrasing causal explanations in terms of pathways paves the way for several types of consistency checks. First, Lemma 4.2 and Theorem 4.3 show how to test consistency *with data* from the "normal regime" because this enables the estimation of the respective probabilities $P$. Second, for any given agent generating causal explanations (human experts or GenAI tools like LLMs), we can check internal consistency, both in a qualitative and in a quantitative sense: For the qualitative test of a pathway $\mathcal{P}$, we can ask the agent about substructures. If $\mathcal{P}$ contains the edge $B_i \to B_j$, we can ask "is $B_i$ a cause of $B_j$ in the context of $\mathbf{B}_{\mathrm{Pa}(j)\setminus\{i\}}$?". For a quantitative test, we can ask about the probability that $B_i = 1$ results in $B_j = 1$, given $\mathbf{B}_{\mathrm{Pa}(j)\setminus\{i\}} = 1$. In other words, we check whether the claimed pathway is consistent with believed probabilities, regardless of whether these are Bayesian beliefs or whether they are inferred from data. We asked LLMs to describe realistic hypothetical causal pathways that explain why an imaginary person in the USA became homeless. We describe more details in Appendix H.

We obtained, among others, a pathway $A \to B \to C \to D \to E$, with the following events:
A. 35-year-old male developed schizophrenia at age 19; lacks consistent access to psychiatric medication due to gaps in Medicaid coverage
B. Fired from three consecutive jobs over 2 years due to paranoid behavior and absences during acute episodes
C. With no income and \$2,000 in savings depleted on food/medication, evicted from rental apartment
D. No family support available—parents died; only sibling cut contact after threatening incident during psychotic break
E. Living on streets for 18+ months; chronic homelessness now established

We then asked the same LLM, in separate conversations, about the probability that each child node was caused by its parent nodes and obtained the following[6] results: $P(B \mid A) \approx 0.55$, $P(C \mid B) \approx 0.8$, $P(D \mid C) \approx 0.05$, $P(E \mid D) \approx 0.2$. Further, the a-priori probability of being homeless long-term was estimated at $P(E) \approx 0.0005$. Taking $A$ as root cause results in a pathway explanation score

---

[6]Evaluating the accuracy of these probabilities is beyond the scope of this paper; in future agentic applications, they could be inferred directly from data.

of only

$$\mathcal{E}_{R \to t}^K = 1 - \frac{\log(0.55 \cdot 0.8 \cdot 0.05 \cdot 0.2)}{\log 0.0005} \approx 1 - \frac{5.43}{7.60} \approx 0.29,$$

since the explanation is mainly weak at the link $C \to D$. After all, $C$ does not refer to any mental illness that would explain event $D$. The evaluation suggests to revise the pathway or events in a way that parent events contain the essential information needed to explain the child event, e.g., by drawing an additional direct link $A \to D$ or rephrasing $C$. Looking at the LLM output in Appendix H.1, the reader may notice that it is inconsistent regarding the hypothesized pathway: the displayed chain suggests the pathway $A \to B \to C \to D \to E$, whereas at the end it specifies edges which contain no arrow $C \to D$, but include an additional link $D \to E$. We also checked that this alternate pathway yields a higher explanation score, aligning with the fact that it avoids the weak link $C \to D$.

As a disclaimer for this example, we would like to emphasize that high explanation scores are not sufficient to prove that a causal explanation is valid. The score does not verify that the underlying causal DAG is true, or that the hypothetical interventional probabilities computed from the DAG reflect the true ones.

## 6. Discussion

Verbal causal explanations of unexpected events are commonly based on chains of events causing each one after another, although the actual causal relation can be a complex graph. We have introduced a formal definition of causal pathway that allows us to quantify the extent to which such simple explanations are consistent with data. The reason why our framework renders explanations falsifiable is that it requires an explicit commitment on which of the mechanisms in a causal network behaved as usual and which ones did not. We do not claim that our framework already enables automated testing of the plausibility of causal explanations. Instead, the goal is to introduce concepts that enable the discussion of plausibility in a quantitative and data-driven way.

## Impact Statement

There is broad consensus that producing reliable and transparent causal explanations would represent a significant advancement in generative AI. Moreover, a key research challenge is to develop methods to assess whether causal statements generated by AI systems are truthful (Bengio et al., 2025). The present work advances this objective by formalizing causal explanations. We present concepts that facilitate empirical falsification and transparency by representing causal explanations as explicit pathways. Through

this approach, we aim to contribute toward improved automated verification methods for causal explanations, with a specific focus on identifying flawed causal accounts of socially significant events. While the inherent ambiguity of verbal explanations makes formal plausibility assessments difficult, LLMs can be guided to generate explanations conforming to the causal pathway format proposed here. Our example in Section 5 demonstrates as a proof of concept illustrating how probabilities derived from the same LLM can serve to evaluate the plausibility of its own explanations.

## Acknowledgments

We thank William Roy Orchard for carefully reading the paper and providing suggestions for improvement.

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

# A. More Details on Explanation Scores

## A.1. Cluster Explanation Score and Distribution Change

**Lemma A.1** (distribution change attribution). *Let $\delta_\mathbf{1}$ denote the point mass on the event $\mathbf{B} = \mathbf{1}$. Then, for any $i \in K$, the cluster explanation score $\mathcal{E}_{i \to k}$ can be written as the quotient of a conditional and a total relative entropy divergence*

$$\mathcal{E}_{i \to K} = \frac{D_{KL}\left[\delta_\mathbf{1}(B_i \mid \mathbf{B}_{\mathrm{Pa}(i)}) \| P_\mathbf{B}(B_i \mid \mathbf{B}_{\mathrm{Pa}(i)})\right]}{D_{KL}\left[\delta_\mathbf{1}(\mathbf{B}) \| P_\mathbf{B}(\mathbf{B})\right]}, \tag{21}$$

*and thus coincides with distribution change attribution (Budhathoki et al., 2021). In other words, each $\mathcal{E}_{i \to K}$ measures the relative contribution of each mechanism to the distribution change from $P_\mathbf{B}(\mathbf{B})$ to $\delta_\mathbf{1}(\mathbf{B})$.*

The proof is a straightforward calculation using

$$D_{KL}\left[\delta_\mathbf{1}(B_i \mid \mathbf{B}_{\mathrm{Pa}(i)}) \| P_\mathbf{B}(B_i \mid \mathbf{B}_{\mathrm{Pa}(i)})\right] = \sum_{\mathbf{b}_{\mathrm{Pa}(i)}} D_{KL}\left[\delta_\mathbf{1}(B_i \mid \mathbf{b}_{\mathrm{Pa}(i)}) \| P_\mathbf{B}(B_i \mid \mathbf{b}_{\mathrm{Pa}(i)})\right] \delta_\mathbf{1}(\mathbf{b}_{\mathrm{Pa}(i)})$$

$$= -\log P_\mathbf{B}(B_i = 1 \mid \mathbf{B}_{\mathrm{Pa}(i)} = \mathbf{1}).$$

## A.2. Explaining Clusters versus Pathways

**Example A.2** (pathway versus cluster explanation score). Pathway explanation score measures whether the root causes are sufficient to explain the *target*, while cluster explanation score measures whether they sufficiently explain the *cluster*. To see the difference, consider the pathway $X_1 \to X_2 \to X_3$, where $X_1$ is the unique root cause. Assume $P(X_1 = 1) = 10^{-15}$ and $X_2 = X_1$. Further, $P(X_3 = 1 \mid X_2 = 0) = 10^{-3}$ and $P(X_3 = 1 \mid X_2 = 1) = 10^{-5}$, that is, the event $X_1 = 1$ renders $X_3 = 1$ even *less* likely. The cluster explanation score reads $1 - \frac{\log(10^{-5})}{\log(10^{-20})} = 1 - 5/20 = 3/4$, while the pathway explanation score is $1 - \frac{\log(10^{-5})}{\log((1-10^{-15})10^{-3} + 10^{-15}10^{-5})} \approx 1 - 5/3 < 0$. For the–very unlikely–joint event $(X_1 = 1, X_2 = 1, X_3 = 1)$ $X_1 = 1$ is a good explanation since it renders it significantly more likely. However, the target event $X_3 = 1$ is not that unlikely. Therefore, $X_1 = 1$ is a bad explanation, it renders the target even less likely.

## A.3. High Pairwise Explanation Score are Not Sufficient

**Example A.3** (OR and AND gate with rare and less rare events). Given the pathway $B_1 \to B_2 \to B_3$ with

$$
\begin{aligned}
B_1 &= N_1, \\
B_2 &= B_1 \wedge N_2, \\
B_3 &= B_2 \vee N_3,
\end{aligned}
$$

where $N_i$ for $i = 1, 2, 3$ are unobserved binary variables with $p(n_1^1) = q_1$ and $p(n_2^1) = p(n_3^1) = q_2$. Assume $q_1 \ll q_2 = q_3 \ll 1$. Then $B_1 = 1$ explains $B_2 = 1$ with explanation score close to 1 because $\log q_2/(\log q_1 + \log q_2) \approx 0$. Further, $B_2 = 1$ explains $B_3 = 1$ with explanation score 1. Hence, both non-root cause nodes $B_2$ and $B_3$ have explanation score close to 1, but the pathway explanation score of the root cause $B_1 = 1$ for the target $B_3 = 1$ reads

$$\mathcal{E}_{1 \to 3}^K = 1 - \frac{\log P(B_3 = 1 \mid B_2 = 1)P(B_2 = 1 \mid B_1 = 1)}{\log P(B_3 = 1)} < 0.$$

# B. Relation to Probabilities of Necessity and Sufficiency

**Definition B.1** (probability of sufficiency and probability of necessity). Let $X, Y$ be binary variables with states $x^0, x^1$ and $y^0, y^1$, respectively, and let $p$ be the joint probability distribution. The probability of sufficiency (PS) is defined as

$$PS := P(Y_{x^1} = y^1 \mid X = x^0, Y = y^0), \tag{22}$$

that is, the probability that $Y = y^1$ would have happened had $X$ been set to $x^1$, conditional on observing $X = x^0, Y = y^0$. Likewise, the probability of necessity (PN) is defined as

$$PN := P(Y_{x^0} = y^0 \mid X = x^1, Y = y^1), \tag{23}$$

that is, the probability that $Y = y^1$ would not have happened had $X$ been set to $x^0$, conditional on observing $X = x^1$, $Y = y^1$.

As shown by (Tian & Pearl, 2000), see Eq. (45), under monotonicity and unconfoundedness these quantities can be expressed as follows:

**Lemma B.2** (PS and PN)**.** *If the binary $X$ is an unconfounded cause of the binary $Y$ and the relation is monotonic (i.e. $p(y^1 \mid x^1) \geq p(y^1 \mid x^0)$), then*

$$PS = \frac{p(y^1 \mid x^1) - p(y^1)}{p(x^0, y^0)}, \tag{24}$$

$$PN = \frac{p(y^1 \mid x^1) - p(y^1 \mid x^0)}{p(y^1 \mid x^1)}. \tag{25}$$

Despite the impression that explanation score is rather related to sufficiency than necessity, we observe that PS can be rather small even when the explanation score is close to 1. Assume, for instance, that $p(y^1) = 10^{-12}$ and $p(y^1 \mid x^1) = 10^{-2}$. Since the denominator in (24) is close to 1, PS is roughly $1/100$, while explanation score is $5/6$. In contrast, when the target is rare and the explanation score is positive enough, the following lemma provides a lower bound showing that PN is close to 1:

**Lemma B.3** (PN and explanation score)**.** *Let the influence of $X$ on $Y$ be monotonic and unconfounded. Then the probability of necessity satisfies*

$$PN \geq 1 - p(y^1)^{\mathcal{E}(x^1 \to y^1)}. \tag{26}$$

*Proof.* Using (45) in (Tian & Pearl, 2000)

$$PN = \frac{p(y^1 \mid x^1) - p(y^1 \mid x^0)}{p(y^1 \mid x^1)} = 1 - \frac{p(y^1 \mid x^0)}{p(y^1 \mid x^1)} \geq 1 - \frac{p(y^1)}{p(y^1 \mid x^1)} = 1 - p(y^1)^{\mathcal{E}(x^1 \to y^1)},$$

where the last step follows from

$$\mathcal{E}(x^1 \to y^1) = 1 - \frac{\log p(y^1 \mid x^1)}{\log p(y^1)},$$

which holds due to $p(y^1 \mid x^1) = p(y^1 \mid do(x^1))$. $\qquad\square$

To illustrate this result with numbers, assume that we observe an event with outlier score $S(y^1) := -\log p(y^1) = 5$, which corresponds to an observation that is unique among $e^5 \approx 148$ observations. For $\mathcal{E}(x^1 \to y^1) = 1/5$ we obtain $PN \geq 1 - e^{-1} \approx 0.63$, meaning that it is more likely for $X = x^1$ to be necessary than not necessary.

Let us now argue why PS is of minor relevance for our explanations. We are interested in explaining an observed event in which both $x^1, y^1$ occurred, while PS concerns cases in which $X = x^0$ and $Y = y^0$ occurred, and asks whether setting $X$ to $x^1$ would have produced $Y = y^1$. Thus, PS addresses a different question from the event-specific explanations considered here. This aligns with an argument of Beckers (2021) on page 7, that "... an explicit sufficiency condition is not required" when focusing on cases where both cause and effect happened. To understand why PN is so close to 1 even when explanation scores are only slightly above 0, note that $Y = y^1$ is unlikely and thus every sufficient cause is unlikely. Thus, although other sufficient causes may exist, they are unlikely to have occurred in the particular occurrence of $Y = y^1$ where $X = x^1$ is also observed. This implies that, in the rare-event setting, a reasonably strong explanatory cause provides evidence for necessity.

It is therefore reasonable to search for causes with high explanation score, because such causes are likely to be necessary for the target event and therefore suitable intervention points for preventing the event, especially when the target is undesirable, such as a performance issue in a technical system. For a pathway with nodes $B_1, \ldots, B_k$ where $B_t$ is the target, we can make this intervention interpretation precise by considering an augmented graph $\mathcal{P}^I$ containing an intervention node $I$ with states $i^0, i^1$. The state $I = i^1$ triggers the intervention $do(\mathbf{B}_R = 1)$ for $I = i^1$ with $P(i^1) =: \epsilon$. Let $P^\epsilon(I, \mathbf{B})$ denote the joint distribution in the augmented DAG. Using Lemma B.3 with $y^1$ being the event $B_t = 1$ and $x^1$ the intervention $i^1$, we conclude that the probability of necessity of $I = i^1$ for $B_t = 1$ satisfies

$$PN \geq 1 - P^\epsilon(B_t = 1)^{\mathcal{E}_{R \to t}}. \tag{27}$$

For $\epsilon \to 0$, $P^\epsilon(\mathbf{B})$ converges to $P(\mathbf{B})$ and $P^\epsilon(B_t = 1)$ can be replaced with $P(B_t = 1)$.

## C. Why We Use Interventional Probabilities Only

One may ask why our explanations use interventional probabilities only –rung 2 in the ladder of causation (Pearl & Mackenzie, 2018)– while reasoning about single events may suggest using counterfactuals (rung 3). To this end, we consider the toy scenario with two binary variables encoding rare events $X, Y$ with $Y = X \wedge N$, where $N$ is binary noise with $P(N = 1) \ll P(X = 1)$. While the counterfactual statement "$Y = 1$ would not have occurred if $X$ had been set to 0 (for that particular statistical unit)" suggests $X$ as crucial for explaining $Y = 1$, it is of minor relevance according to the quantitative measures used here because the counterfactual relevance generalizes to only a tiny fraction of statistical units. However, for the vast majority of units, $X$ has no effect on $Y$, since $N = 0$ is overwhelmingly more likely than $N = 1$. While we do not in general deny the importance of counterfactuals for understanding singular events, our goal of stating explanations that are easily falsifiable and generalize to a significant fraction of statistical units lets us prefer referring to interventional probabilities only.

## D. Comparison to Other Methods for Explaining Rare Events

To provide a concrete illustration with mean-based attribution, consider the logical AND gate $Y = X \wedge N$ with $P(X = 1) = 10^{-9}$ and $P(N = 1) = 10^{-1}$, so that $P(Y = 1) = 10^{-10}$. With our logarithmic scaling (following explanation scores), the contribution of $X$ reads $1 - \frac{1}{10} = 90\%$, in other words its contribution is strong because it increases the probability of the output event by a large factor. By contrast, a mean-based contribution such as $\frac{\mathbb{E}[Y|do(X=1)] - \mathbb{E}[Y]}{1 - \mathbb{E}[Y]}$ is approximately $10^{-1}$, which is much smaller. If we instead consider Shapley-based contributions, the contribution of $X$ is

$$\frac{\mathbb{E}[Y \mid X = 1] - \mathbb{E}[Y] + \mathbb{E}[Y \mid X = 1, N = 1] - \mathbb{E}[Y \mid N = 1]}{2(1 - \mathbb{E}[Y])} \approx 0.55$$

This gives a stronger contribution than the plain mean-based score, but it still remains substantially below the logarithmic score of 0.9, since of the two marginal terms are still tied to the absolute change in the mean. Since we follow arguments in (Oesterle et al., 2025; Budhathoki et al., 2022a) stating that a substantially more rare factor should get significantly more attention, we prefer the logarithmic contribution analysis. Another issue with Shapley-based contributions is that they require knowing the structural causal model and are not invariant under redefining the noise: if we replace $N$ by two variables $N_1, N_2$, with $Y = X \wedge N_1 \wedge N_2$ and $P(N_1 = 1, N_2 = 1) = P(N = 1) = 10^{-1}$, then the Shapley contribution of $X$ changes.

## E. When to Expect Multiple Root Causes

Let us consider a simple causal DAG where $Y$ is an additive effect of multiple independent causes: $Y := \sum_{i=1}^{d} X_i$, with $X_i$ independently identically distributed with density $p_X$. It is known from a large number of asymptotical results, see e.g. (Armendáriz & Loulakis, 2011), that the shape of $p_X$ determines whether extremes of $Y$ tend to originate from extreme values of one of the $X_i$ or multiples of them. It is easy to see why Gaussian $X_i$ asymptotically result in *all* $X_i$ being extreme: since the joint density is rotation invariant in $\mathbb{R}^d$, we can choose $Y$ as one axis, and all directions orthogonal to $Y$ are independent. Accordingly, conditioning on $Y$ being large does not affect the joint distribution of the subspace orthogonal to $Y$. Hence, there are most likely no extreme values in any normalized linear combination of $X_i$ that is orthogonal to $Y$. Therefore *all* $X_i$ must attain large values. On the other hand, Armendáriz & Loulakis (2011) show that for $p_X$ with subexponential density, extreme values of $Y$ originate from extrema of a single $X_i$ in the limit of large $y$. All the above arguments are based on the assumption that the outlier is a result of post-selection, rather than distribution shift.

If we, instead, assume that our event is a result of distribution shift, which is caused by shifts of conditional distributions at multiple nodes, we tend to ask for a common root cause behind distributional shifts happening *simultaneously* at actually "independent mechanisms", cf Peters et al. (2017), Sections 2.1 and 2.2.

## F. Proof of Theorem 4.3

Let $1, \ldots, n$ be w.l.o.g. causally ordered according to $\mathcal{G}$. Define the random variable

$$(x_j, \mathbf{x}_{\mathrm{Pa}(j)}) \mapsto L_j(x_j, \mathbf{x}_{\mathrm{Pa}(j)}) := -\log P_{\mathbf{X}}\big(\tau_j(X_j) \geq \tau_j(x_j) \mid \tau_{\mathrm{Pa}(j)}(\mathbf{X}_{\mathrm{Pa}(j)}) \geq \tau_{\mathrm{Pa}(j)}(\mathbf{x}_{\mathrm{Pa}(j)})\big).$$

Lemma 4.2 entails for all $j \notin R$, given an arbitrary observation $\mathbf{x}_{\{1,\ldots,j-1\}}$, the random variable $L_j$ attains values above $\alpha$ with probability at most $e^{-\alpha}$. Let $(T_j)_{j \in N \setminus R}$ be independent exponentially distributed random variables with density $p(t) = e^{-t}$. Then for any $\mathbf{x}_{\{1,\ldots,j-1\}}$, the conditional distribution of $L_j$ given $\mathbf{x}_{\{1,\ldots,j-1\}}$, is stochastically dominated by $T_j$. Since $(L_i)_{i \in N \setminus R, i \leq j-1}$ is a function of $\mathbf{x}_{\{1,\ldots,j-1\}}$, we also conclude that $L_j$ is stochastically dominated by $T_j$, given any values of $(L_i)_{i \in N \setminus R, i \leq j-1}$. Hence, $L = \sum_{j \in N \setminus R} L_j$ is stochastically dominated by $\sum_{j \in N \setminus R} T_j$.

The sum of $n'$ independent identically distributed exponential distributions with parameter $\lambda$ is the Erlang distribution $\mathrm{Erl}(\lambda, n')$ (Ibe, 2013), with CDF

$$F(x) = 1 - e^{-\lambda x} \sum_{i=0}^{n'-1} \frac{(\lambda x)^i}{i!}.$$

Since $\lambda = 1$ because we have density $p(t) = e^{-t}$ and $n' = n - |R|$, we conclude that $T$ attains values greater than or equal to some $c > 0$ with probability at most

$$p = \sum_{i=0}^{n-|R|-1} \frac{c^i}{i!} e^{-c}, \tag{28}$$

which applies to $L$ too because it is stochastically dominated by $T$.

## G. Additional Results on Pathway Abstractions

### G.1. Further Analysis of Example 4.8

We first consider the simpler pathway abstraction $B_2 \rightarrow B_3$, where both the pathway and the cluster are given by the DAG $B_2 \rightarrow B_3$, and we set $P_{\mathbf{B}}(B_2, B_3) := P(B_2, B_3)$. The accuracy of the bivariate abstraction is

$$r_{\mathrm{bi}} = 1 - \frac{\max\{D_{KL}(\mathrm{Bern}(p_0) \,\|\, \mathrm{Bern}(q_0)), D_{KL}(\mathrm{Bern}(p_1) \,\|\, \mathrm{Bern}(q_1))\}}{-\log P(Z \geq z)},$$

where $q_0$ and $q_1$ are computed by

$$q_0 := P_{\mathbf{B}}(B_3 = 1 \mid do(B_2 = 0)) = P_{\mathbf{B}}(B_3 = 1 \mid B_2 = 0) = P(Z \geq z \mid Y < y), \tag{29}$$

$$q_1 := P_{\mathbf{B}}(B_3 = 1 \mid do(B_2 = 1)) = P_{\mathbf{B}}(B_3 = 1 \mid B_2 = 1) = P(Z \geq z \mid Y \geq y), \tag{30}$$

while $p_0$ and $p_1$ are obtained from the fine-grained interventional probability distributions under natural micro-realization given by

$$p_0 := P(B_3 = 1 \mid do(B_2 = 0)) = \int_{-\infty}^{y} P(Z \geq z \mid do(Y = y'))p(y' \mid Y < y)dy', \tag{31}$$

$$p_1 := P(B_3 = 1 \mid do(B_2 = 1)) = \int_{y}^{\infty} P(Z \geq z \mid do(Y = y'))p(y' \mid Y \geq y)dy'. \tag{32}$$

Since $Y$ and $Z$ are linear functions of the jointly Gaussian variables $X, N_Y, N_Z$, the pair $(Y, Z)$ is bivariate Gaussian with covariance

$$\Sigma_{Y,Z} = \begin{pmatrix} 1 & \gamma + \alpha\beta \\ \gamma + \alpha\beta & 1 \end{pmatrix},$$

and therefore the probabilities in (29) and (30) can be computed from the bivariate Gaussian CDF of $(Y, Z)$. To compute the interventional probability $P(Z \geq z \mid do(Y = y'))$, consider the structural equation model

$$Z^* = \beta X + \gamma Y^* + N_Z, \tag{33}$$

where $X$ and $N_z$ are as in the original structural model in (20), and $Y^* \sim \mathcal{N}(0, 1)$ with $Y^* \perp X, N_z$.

Then, conditional on $Y^* = y'$, we have

$$Z^* = \beta X + \gamma y' + N_Z.$$

Hence,

$$P(Z \geq z \mid do(Y = y')) = P(Z^* \geq z \mid Y^* = y').$$

Therefore,

$$P(Z \geq z \mid do(B_2 = 1)) = \int_y^\infty P(Z^* \geq z \mid Y^* = y')p(y' \mid Y^* \geq y')dy = P(Z^* \geq z \mid Y^* \geq y'). \tag{34}$$

Similarly we obtain

$$P(Z \geq z \mid do(B_2 = 0)) = P(Z^* \geq z \mid Y^* < y'). \tag{35}$$

Note that $(Y^*, Z^*)$ is a bivariate Gaussian variable with covariance

$$\Sigma_{Y^*,Z^*} = \begin{pmatrix} 1 & \gamma \\ \gamma & \beta^2 + \gamma^2 + \sigma_{N_z}^2 \end{pmatrix},$$

since $\mathrm{Cov}(Y^*, Z^*) = \gamma$ and $\mathrm{Var}(Z^*) = \beta^2 + \gamma^2 + \sigma_z^2$. Thus, the right-hand sides of (34) and (35) can be calculated using the bivariate Gaussian CDF of $(Y^*, Z^*)$.

The pathway explanation score of the bivariate pathway abstraction is given by

$$\mathcal{E}_{2 \to 3}^{\{2,3\}} = 1 - \frac{\log P(Z \geq z \mid Y \geq y)}{\log P(Z \geq z)}.$$

Now we consider the pathway abstraction $B_1 \to B_3 \leftarrow B_2$, with the full DAG as the cluster and $P_\mathbf{B}(\mathbf{B}) := P(\mathbf{B})$. For the trivariate abstraction, the accuracy is

$$r_{tri} = 1 - \frac{\max_{S \subseteq \{1,2,3\}} \max_{\mathbf{b}_S} D_{KL}(P(\mathbf{B} \mid do(\mathbf{B}_S = \mathbf{b}_S)) \,\|\, P_\mathbf{B}(\mathbf{B} \mid do(\mathbf{B}_S = \mathbf{b}_S)))}{-\log P(Z \geq z)}.$$

Since $P_\mathbf{B}$ is Markov relative to the cluster, we have the factorization

$$p_\mathbf{B}(b_1, b_2, b_3) = p_\mathbf{B}(b_1)p_\mathbf{B}(b_2 \mid b_1)p_\mathbf{B}(b_3 \mid b_1, b_2).$$

Thus, $P_\mathbf{B}(\mathbf{B} \mid do(\mathbf{B}_s = \mathbf{b}_s))$ is obtained by fixing the variables in $S$ and multiplying only the mechanisms of the non-intervened variables.

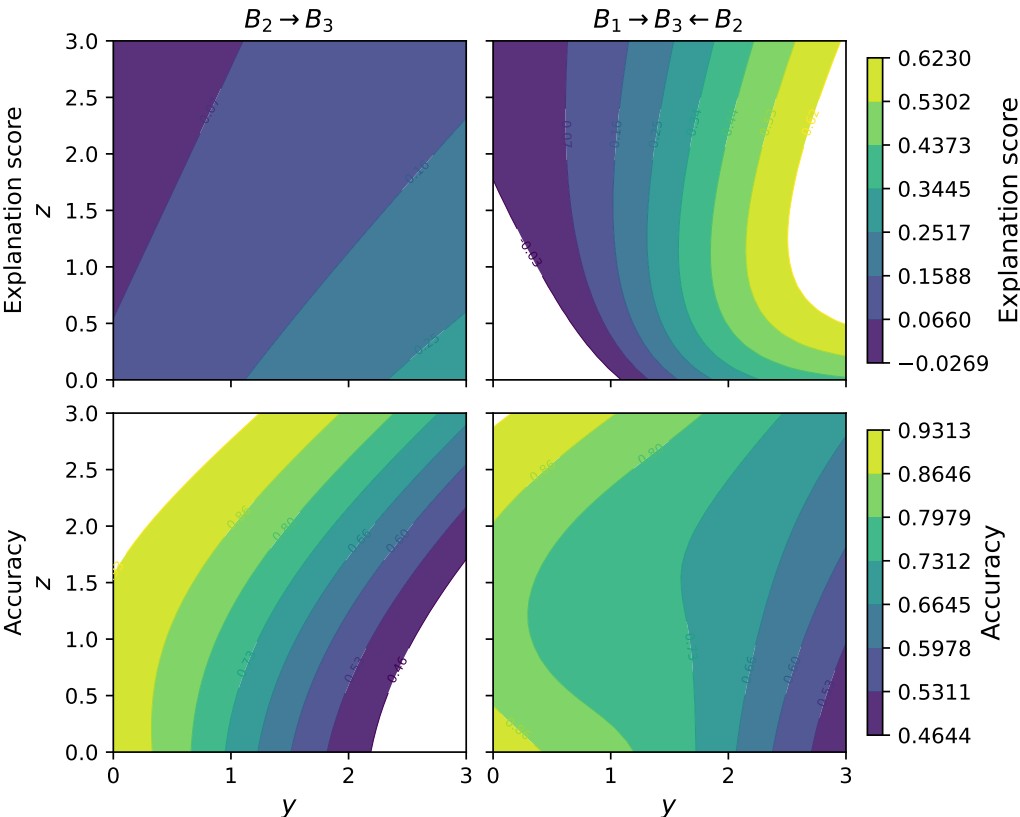

*Figure 4.* Explanation scores and abstraction accuracies for Example 4.8, comparing the bivariate abstraction $B_2 \rightarrow B_3$ with the trivariate abstraction $B_1 \rightarrow B_3 \leftarrow B_2$. In all figures $x = 1$, and parameters are $\alpha = -0.9$, $\beta = 0.9$, and $\gamma = 0.9$. The trivariate abstraction achieves higher explanation scores and accuracies in the region $y \geq 2$, $z \geq 2$ because $B_1$ captures contextual information that is lost in the bivariate abstraction.

We now describe the fine-grained interventional distributions $P(\mathbf{B} \mid do(\mathbf{B}_s = \mathbf{b}_s))$. Interventions that do not involve $B_2$ can be simply computed using the observational Gaussian distribution of $(X, Y, Z)$. For example, under $do(B_2 = 1)$, the fine-grained value of $X$ is drawn from $P(X \mid |X| \leq x)$, and $Y$ and $Z$ are generated by the original structural equation model. Hence,

$$P(B_1 = 1, B_2 = 1, B_3 = 1 \mid do(B_1 = 1)) = P(Y \geq y, Z \geq z \mid |X| \leq x).$$

When $B_2$ is intervened on, we again use structural equation model in (33). Then the covariance matrix of the multivariate Gaussian variable $(X, Y^*, Z^*)$ is given by

$$\Sigma_{X,Y^*,Z^*} = \begin{pmatrix} 1 & 0 & \beta \\ 0 & 1 & \gamma \\ \beta & \gamma & \beta^2 + \gamma^2 + \sigma^2_{N_Z} \end{pmatrix},$$

The fine-grained interventional probabilities can therefore be calculated using Gaussian probabilities of the multivariate Gaussian variable $(X, Y^*, Z^*)$. For example, under $do(B_2 = 1)$,

$$P(B_1 = 1, B_2 = 1, B_3 = 1 \mid do(B_2 = 1)) = \frac{P(|X| \leq x, Y^* \geq y, Z^* \geq z)}{P(Y^* \geq y)},$$

and under $do(B_1 = 1, B_2 = 1)$ we have

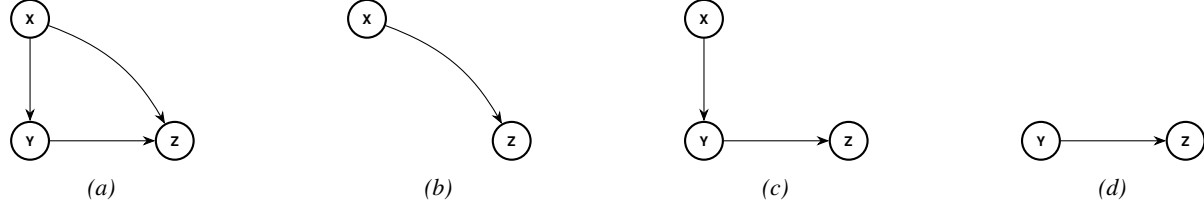

*Figure 5.* (a) Causal DAG with binary variables $X$, $Y$, and $Z$, where $X$ influences the target $Z$ directly and indirectly via the mediator $Y$. (b): Pathway abstraction with accuracy 1, which is always possible because the indirect and direct links can always be merged into one direct link. (c): Pathway abstraction whose accuracy $r$ depends on the relevance of the direct link $X \to Y$ for the outlier event at $Z$ (d): Pathway abstraction whose accuracy $r$ depends on the relevance of the confounder $X$ for the outlier event at $Z$.

$$P(B_1 = 1, B_2 = 1, B_3 = 1 \mid do(B_1 = 1, B_2 = 1)) = \frac{P(|X| \le x, Y^* \ge y, Z^* \ge z)}{P(|X| \le x)P(Y^* \ge y)}.$$

Figure 4 compares the accuracy and explanation score of the bivariate abstraction $B_2 \to B_3$ with the trivariate abstraction $B_1 \to B_3 \leftarrow B_2$ in Example 4.8 for $x = 1$ and different values of $y, z$. The parameter choice creates negative confounding: because $\alpha < 0$, large $Y$ tends to occur with smaller values of $X$, while because $\beta > 0$, smaller values of $X$ tend to decrease $Z$. The trivariate abstraction includes $B_1$, which captures this context, and therefore achieves higher explanation scores and accuracies in the outlier region $y \ge 2, z \ge 2$.

The code used for the numerical evaluation in Example 4.8 and for generating the plots in Figure 4 is available at https://github.com/AH-20/CausalPathwayExplanation.

### G.2. Pathway Abstraction of DAGs With Binary Variables

Consider the causal DAG given by Figure 5a, where $X, Y, Z$ are binary variables, and the events $X = i, Y = j, Z = k$ are denoted by $x^i, y^j, z^k$, respectively, for $i, j, k \in \{0, 1\}$. Note that the DAG in Figure 5b is always a valid pathway abstraction with accuracy equal to 1 by merging the direct and indirect links into one arrow. In this section we show that the accuracy of the pathway abstraction in Figure 5c depends on the relevance of the direct link $X \to Z$ and we derive a quantitative condition for safely ignoring the confounder $X$ to obtain the pathway abstraction in Figure 5d.

First, in close analogy to Budhathoki et al. (2022a), we define a notion of "causal contribution" for a cause effect-pair $X \to Y$:

**Definition G.1** (causal contribution). For $i, j \in \{0, 1\}$, the causal contribution of $x^i$ to $y^j$ is

$$C\left(x^i \to y^j\right) := \log \frac{p(y^j \mid x^i)}{p(y^j)}. \tag{36}$$

Note that it coincides with explanation score up to a normalization factor:

$$\mathcal{E}\left(x^i \to y^j\right) = -\frac{C\left(x^i \to y^j\right)}{\log p(y^j)}.$$

When there are mediator nodes between the cause-effect pair in addition to the direct link, the above definition does not correctly capture the "direct" contribution of the cause node to the effect node. Therefore, we introduce a notion of direct contribution that follows our principle of logarithmic contribution analysis, although it is motivated by the average controlled direct effect in Definition 3 in (Pearl, 2001).

**Definition G.2** (controlled direct contribution). For $i, j, k \in \{0, 1\}$, the controlled direct contribution of $x^i$ to $z^k$ where $Y$ is set to $y^j$ is

$$C^c_{\text{direct}}\left(x^i \to z^k \mid y^j\right) := \log \frac{p(z^k \mid do(x^i, y^j))}{p(z^k \mid do(y^j))}. \tag{37}$$

We are now able to describe a condition for which we can safely ignore the confounder:

**Lemma G.3** (ignoring confounders). *Let $X, Y, Z$ be binary variables connected by the DAG $\mathcal{G}$ in Figure 5a, with joint probability distribution $P$. Consider the pathway abstraction in Figure 5d, with root cause $Y$ and target $Z$. We choose the cluster DAG to coincide with this pathway DAG, and set $B := \{Y, Z\}$, $p_B(y^j, z^k) := p(y^j, z^k)$. Then this pathway abstraction has accuracy at least*

$$r \geq 1 - \frac{s_{\text{confounding}}}{-\log p(z^1)}, \tag{38}$$

*where $s_{\text{confounding}}$ is a parameter measuring the strength of confounding by $X$, which is defined by*

$$s_{\text{confounding}} := \max_{j,k} \min \left\{ \max_i \left| C\left(x^i \to y^j\right) \right|, \left| C^c_{\text{direct}}\left(x^1 \to z^k \mid y^j\right) - C^c_{\text{direct}}\left(x^0 \to z^k \mid y^j\right) \right| \right\}. \tag{39}$$

*Proof.* Since $Y$ and $Z$ are already binary, it follows from definition 4.4 that the natural micro-realization coincides with ordinary binary interventions on the abstract variables. Moreover, since $P_B(Y, Z) = P(Y, Z)$, the observational distribution is matched by construction. Interventions on $Z$, or on both $Y$ and $Z$, also agree in the original and abstract models because $Z$ is fixed by the intervention and the marginal distribution of $Y$ is unchanged. Therefore, to find the accuracy of the pathway abstraction, we only need to compute the following:

$$r = 1 - \max_j \left\{ \frac{D_{KL}\left(P(Z \mid do(y^j)) \,\|\, P_B(Z \mid do(y^j))\right)}{-\log p(z^1)} \right\} = 1 - \max_j \left\{ \frac{\sum_k p(z^k \mid do(y^j)) \log \frac{p(z^k \mid do(y^j))}{p_B(z^k \mid do(y^j))}}{-\log p(z^1)} \right\}. \tag{40}$$

We can write

$$\log \frac{p_B(z^k \mid do(y^j))}{p(z^k \mid do(y^j))} = \log \frac{p(z^k \mid y^j)}{p(z^k \mid do(y^j))} = \log \frac{\sum_i p(z^k \mid x^i, y^j) p(x^i \mid y^j)}{\sum_i p(z^k \mid x^i, y^j) p(x^i)}. \tag{41}$$

Since the numerator and denominator of (41) are two different convex sums of $p(z^k \mid x^1, y^j)$ and $p(z^k \mid x^0, y^j)$, we have

$$\left| \log \frac{p_B(z^k \mid do(y^j))}{p(z^k \mid do(y^j))} \right| \leq \left| \log \frac{p(z^k \mid x^1, y^j)}{p(z^k \mid x^0, y^j)} \right| = \left| \log \frac{p(z^k \mid x^1, y^j)}{p(z^k \mid do(y^j))} - \log \frac{p(z^k \mid x^0, y^j)}{p(z^k \mid do(y^j))} \right|$$
$$= \left| C^c_{\text{direct}}\left(x^1 \to z^k \mid y^j\right) - C^c_{\text{direct}}\left(x^0 \to z^k \mid y^j\right) \right|. \tag{42}$$

Further, we have

$$\log \frac{\sum_i p\left(z^k \mid x^i, y^j\right) p(x^i \mid y^j)}{\sum_i p(z^k \mid x^i, y^j) p(x^i)} \in \left[ \min_i \log \frac{p(x^i \mid y^j)}{p(x^i)}, \max_i \log \frac{p(x^i \mid y^j)}{p(x^i)} \right],$$

because

$$\frac{\sum_m \lambda_m a_m}{\sum_m \lambda_m b_m} \in \left[ \min_m \frac{a_m}{b_m}, \max_m \frac{a_m}{b_m} \right],$$

for any positive sequences $\lambda_m, a_m, b_m$.

With $p(x^i \mid y^j)/p(x^i) = p(y^j \mid x^i)/p(y^j)$ we have thus shown that the absolute value of (41) is bounded by

$$\max_i \left\{ \left| \log \frac{p(y^j \mid x^i)}{p(y^j)} \right| \right\} = \max_i \left\{ \left| C\left(x^i \to y^j\right) \right| \right\}. \tag{43}$$

Since the KL-divergence in (40) is a convex sum of the log-ratios in (41), from (42) and (43) we have

$$D_{KL}\left(P(Z \mid do(y^j)) \,\|\, P_B(Z \mid do(y^j))\right)$$
$$\leq \max_{j,k} \min \left\{ \max_i \left| C\left(x^i \to y^j\right) \right|, \left| C^c_{\text{direct}}\left(x^1 \to z^k \mid y^j\right) - C^c_{\text{direct}}(x^0 \to z^k \mid y^j) \right| \right\}, \tag{44}$$

which completes the proof.

$\square$

Lemma G.3 shows that $X$ can be ignored when the confounding strength is small. This happens, for example, if $X$ has little contribution to $Y$, or if the controlled direct contribution of $X$ to $Z$ changes little across the values of $X$ once $Y$ is fixed.

Finally, we are able to describe a condition for which we can safely ignore the direct link:

**Lemma G.4** (ignoring direct link). *Let $X, Y, Z$ be binary variables connected by the DAG $\mathcal{G}$ in Figure 5a with joint probability distribution $P$. Consider the pathway abstraction in Figure 5c, with root cause $X$ and target $Z$. We choose the cluster DAG to coincide with this pathway DAG and set $\mathbf{B} := \{X, Y, Z\}$, $p_{\mathbf{B}}(x^i, y^j, z^k) := p(x^i, y^j)p(z^k \mid y^j)$. Let*

$$M := \max_{i,j,k} \left| C_{\text{direct}}^c \left( x^i \to z^k \mid y^j \right) \right|, \tag{45}$$

*then, the pathway abstraction has accuracy at least*

$$r \geq 1 - \frac{2M}{-\log p(z^1)}. \tag{46}$$

*Proof.* Since $X, Y, Z$ are already binary, the natural micro-realization coincides with ordinary binary interventions. By definition of the pathway abstraction, $p_{\mathbf{B}}(x^i, y^j, z^k) = p(x^i, y^j)p(z^k|y^j)$, so the mechanisms for $X$ and $Y$ coincide with those of the original model. Therefore, if $Z$ is intervened on, the KL-divergence is zero, so we only need to consider intervention sets $\mathbf{B}_S$ with $Z \notin \mathbf{B}_S$, which yields

$$D_{KL}\left(P(\mathbf{B} \mid do(\mathbf{B}_S = \mathbf{b}_S)) \| P_{\mathbf{B}}(\mathbf{B} \mid do(\mathbf{B}_S = \mathbf{b}_s))\right) = \sum_{\mathbf{b}} P(\mathbf{b} \mid do(\mathbf{B}_S = \mathbf{b}_S)) \log \frac{p(z^k|x^i, y^j)}{p(z^k|y^j)} \tag{47}$$

where the sum is over configurations $\mathbf{b} = (x^i, y^j, z^k)$ consistent with the intervention. Therefore, we need to bound

$$\left| \log \frac{p(z^k|x^i, y^j)}{p(z^k|y^j)} \right|.$$

Note that since $p(z^k|do(x^i, y^j)) = p(z^k|x^i, y^j)$, we have

$$\left| \log \frac{p(z^k \mid x^i, y^j)}{p(z^k \mid do(y^j))} \right| \leq M,$$

therefore

$$e^{-M} p(z^k \mid do(y^j)) \leq p(z^k \mid x^i, y^j) \leq e^M p(z^k \mid do(y^j)). \tag{48}$$

Since

$$p(z^k \mid y^j) = \sum_i p(z^k \mid x^i, y^j)p(x^i \mid y^j),$$

is a convex sum of the probabilities $p(z^k \mid x^i, y^j)$, from (48) we have

$$e^{-M} p(z^k \mid do(y^j)) \le p(z^k \mid y^j) \le e^M p(z^k \mid do(y^j)), \tag{49}$$

then from (48) and (49) it follows that

$$e^{-2M} \le \frac{p(z^k \mid x^i, y^j)}{p(z^k \mid y^j)} \le e^{2M}.$$

Therefore

$$\left| \log \frac{p(z^k \mid x^i, y^j)}{p(z^k \mid y^j)} \right| \le 2M.$$

and thus

$$\max_{S, \mathbf{b}_S} D_{KL} \left( P(\mathbf{B} \mid do(\mathbf{B}_S = \mathbf{b}_S)) \,\|\, P_{\mathbf{B}}(\mathbf{B} \mid do(\mathbf{B}_S = \mathbf{b}_S)) \right) \le 2M,$$

which completes the proof.

$\square$

Lemma G.4 shows that the direct edge $X \to Y$ can be ignored when the controlled direct contribution of $X$ to $Y$ is uniformly small across all values of $X, Y, Z$.

## H. Example of a Causal Pathway Generated by LLM

Generated by Anthropic Claude Opus 4.5, with 32,000 output tokens, accessed via Amazon Bedrock. The boxes contain answers in a format that is close to the original output.

### H.1. Generation of an Example with Pathway

---

**System prompt: describe causal pathway**

**User input:**   Describe a causal pathway that explains how an imaginary inhabitant of the USA got long-term homeless. The pathway should be a causal DAG where each node is a condition describing an objective fact. Every condition (that is, node of the DAG) is supposed to be a likely consequence of its parents, that is, they are explained by the parent conditions. The description of the parent conditions need to be detailed enough such that the consequence is likely given the parent conditions, without referring to remote ancestors. Only the root causes are one or more nodes (not necessarily the root nodes of the DAG) describe conditions which need not be explained in terms of their parents. Please be explicit about the nodes and the arrows of the DAG and try to restrict the explanation to a DAG with not more than 5 nodes.

**LLM output:**   (Readers may realize that the description of the pathway below is logically inconsistent with the list of arrows further below. We nevertheless considered this pathway in Section 5 because it provided an instructive example of an incomplete explanation. Since the scope of this paper is not the evaluation of LLMs, we believe it is justified to pick examples that are particularly suitable for our purpose.)

\# Causal Pathway to Long-Term Homelessness
\#\# DAG Structure
"'

---

[A] Severe Mental Illness (Schizophrenia) diagnosed at age 19

    ↓

[B] Unable to maintain employment due to untreated psychotic episodes

    ↓

[C] Exhausted savings and lost apartment after 8 months without income

    ↓

[D] Family estrangement (parents deceased, sibling relationship severed during psychotic episode)

    ↓

[E] Long-term homelessness (>1 year on streets) "'

## Nodes

| Node | Condition |
|------|-----------|
| A | 35-year-old male developed schizophrenia at age 19; lacks consistent access to psychiatric medication due to gaps in Medicaid coverage. |
| B | Fired from three consecutive jobs over 2 years due to paranoid behavior and absences during acute episodes. |
| C | With no income and $2,000 in savings depleted on food and medication, evicted from rental apartment. |
| D | No family support available—parents deceased; only sibling cut contact after a threatening incident during a psychotic break. |
| E | Living on the streets for more than 18 months; chronic homelessness now established. |

## Root Cause
**Node A** (Severe Mental Illness) is the **root cause**—a condition that initiates the cascade but is not explained by other nodes in this pathway.
## Arrows
- A → B
- B → C
- C → E
- D → E (D combines with C to make E likely)

## H.2. Inferring Probability $P(B|A)$

**System prompt: estimate a probability**

**User input:** what's the probability that condition A results in condition B for an inhabitant in the US? | A | 35-year-old male developed schizophrenia at age 19; lacks consistent access to psychiatric medication due to gaps in Medicaid coverage | | B | Fired from three consecutive jobs over 2 years due to paranoid behavior and absences during acute episodes |

**LLM output:** Model: # Probability Estimate: Condition A → Condition B
## My Estimate: **45-65%** (moderate-to-high probability)

—

## Key Factors Increasing Probability
### Schizophrenia + Medication Non-Adherence
- **Relapse rates** without consistent medication: 60-80% within 1 year
- Acute episodes typically involve positive symptoms (paranoia, disorganization) that are highly visible in workplace settings
### Employment Vulnerability
- Schizophrenia baseline unemployment rate: ∼70-80%
- Among those who *do* work, job tenure is often short

- Paranoid subtype symptoms are particularly problematic for workplace relationships
### Medicaid Gap Impact
- Coverage disruptions strongly correlate with psychiatric hospitalization and crisis episodes
- 16 years of illness duration suggests multiple cycles of decompensation
—

## Factors That Could Lower Probability

| Factor | Effect |
|--------|--------|
| Strong family support system | May buffer crises |
| Milder symptom severity | Some maintain function even unmedicated |
| Supportive employment programs | If accessed, improve retention |
| Not all jobs terminate *for cause* | Some separations may be voluntary |

## Why Not Higher ($> 65\%$)?
- Not all people with poorly-controlled schizophrenia hold three jobs in 2 years (many hold zero)
- The specific pattern of "three firings" requires repeated re-hiring, which suggests *some* baseline functionality

—

**Point estimate: $\sim 55\%$**

## H.3. Inferring probability $P(C|B)$

---

**System prompt: estimate a probability**

**User input:** What's the probability that [B] Unable to maintain employment due to untreated psychotic episodes results in [C] Exhausted savings and lost apartment after 8 months without income for an inhabitant of the USA?

**LLM output:** Probability Estimate: B $\rightarrow$ C
Scenario Analysis
Given: A person in the USA who is unable to maintain employment due to untreated psychotic episodes
Question: Probability of exhausting savings and losing apartment after 8 months without income
Key Factors
Factors Increasing Probability:

- Limited savings baseline: ˜56% of Americans cannot cover a $1,000 emergency; median savings for renters is very low

- 8 months is substantial: Average U.S. rent is ˜$1,500-2,000/month = $12,000-16,000 needed just for housing

- Untreated psychosis context: Suggests limited access to support systems, possible prior instability

- Eviction timelines: Most states allow eviction proceedings after 1-3 months of non-payment

Factors Decreasing Probability:

- Disability benefits: SSI/SSDI applications (though these take 6+ months typically)

- Family support: Some may have family assistance

- Emergency rental assistance programs

- Savings variation: Some individuals do have substantial savings

Estimate

Given:

- Most Americans lack 8 months of expense coverage

- Untreated psychosis population likely has *below-average* financial reserves

- Eviction processes typically complete within 8 months of non-payment

Probability: 0.75 - 0.85

Point estimate: $\sim 0.80$

## H.4. Inferring Probability $P(D|C)$

**System prompt: estimate a probability**

**User input:**  What's the probability that [C] Exhausted savings and lost apartment after 8 months without income results in [D] Family estrangement (parents deceased, sibling relationship severed during psychotic episode) for an inhabitant of the USA?

**LLM output:**  I need to estimate the probability that exhausting savings and losing an apartment after 8 months without income leads to family estrangement (with parents deceased and sibling relationship severed during a psychotic episode).
Key considerations:

1. Baseline context: Someone who has gone 8 months without income and lost housing is likely experiencing severe crisis - possibly involving mental health issues, substance abuse, or other destabilizing factors.

2. The psychotic episode element: This is specific - it implies a serious mental health condition. Psychotic episodes can involve behavior that severely damages relationships (paranoid accusations, erratic actions, inability to recognize loved ones' intentions).

3. Conditional factors:

   - The housing loss suggests someone who didn't have family support to fall back on - this may indicate pre-existing strain
   - 8 months without income suggests possible disability, mental health crisis, or other severe impairment
   - Sibling relationships severed "during" a psychotic episode suggests the break happens in crisis, not gradually

4. However: Most people who lose housing don't have psychotic episodes, and most psychotic episodes don't permanently sever all family ties. This is a specific tragic pathway.

Estimation:

- The scenario C already selects for vulnerable individuals

- But complete family estrangement with this specific pattern remains relatively uncommon

- Partial estrangement is more common than total severance

Probability estimate: 3-7%

Point estimate: ~0.05 (5%)

**H.5. Inferring Probability** $P(E|D)$

---

**System prompt: estimate a probability**

**User input:** What's the probability that [D] Family estrangement (parents deceased, sibling relationship severed during psychotic episode) results in [E] Long-term homelessness ($>$1 year on streets) for an inhabitant of the USA?

**LLM Output:** Estimating P(Long-term Homelessness | Family Estrangement with Psychotic Episode)
Key Factors to Consider
Population context: - Person has experienced a psychotic episode (suggesting serious mental illness) - Complete loss of family safety net (parents deceased, sibling estrangement) - Located in USA
Breaking Down the Estimate
Baseline rates: - $\sim$30% of homeless individuals have serious mental illness - $\sim$50-60% of long-term homeless cite family conflict as contributing factor - Average duration of homelessness is longer for those with SMI
Protective vs. Risk factors in this scenario:
Risk Factors: Psychotic episode history; No family support; Relationship severed during episode
Potential Mitigators: Access to mental health systems; Disability benefits (SSI/SSDI); Possible reconnection after stabilization; Other social supports possible
Probability Estimate
For someone with this specific profile (SMI + complete family estrangement):
Point estimate: 15-25%
Reasoning: - Much higher than general population ($\sim$0.2% long-term homeless) - SMI alone increases risk $\sim$10-20$\times$ - Loss of family support roughly doubles risk again - But majority still avoid long-term homelessness through benefits, treatment, other relationships
Central estimate: $\sim$20%
This reflects serious but not deterministic risk.

---

