# OpenReview forum: "Formalizing and Falsifying Causal Pathways of Rare Events"
_ICML.cc/2026/Conference — ICML 2026 regular_

### Official Review · Reviewer_uvjU · 2026-03-11

**Soundness:** 3
**Presentation:** 3
**Significance:** 3
**Originality:** 3
**Overall Recommendation:** 4
**Confidence:** 3

**Summary:**

The paper addresses the critical challenge of event-specific causal explanation, particularly focusing on rare events such as stock market crashes or natural disasters. Traditional Root Cause Analysis (RCA) often focuses on identifying the source but fails to explain the propagation mechanism (the "how"). Moreover, the authors introduce a rigorous mathematical framework centered on the Pathway Explanation Score, which quantifies the degree to which a specific causal subgraph explains a target event using KL-divergence metrics. A key innovation is Pathway Abstraction, a method that bridges the gap between complex continuous variables and interpretable binary event chains, ensuring that causal claims remain grounded in structural model knowledge.

**Compliance With Llm Reviewing Policy:**

Affirmed.

**Final Justification:**

I thank the authors for their detailed rebuttal and clarifications. The paper presents a theoretically rigorous framework for event-specific causal explanation, with strong foundations in structural causal models and probabilistic theory. The proposed Pathway Explanation Score and Pathway Abstraction are novel and offer a meaningful advance over traditional root cause analysis by capturing both causal sources and propagation mechanisms. This contributes significantly to explainable AI, especially in high-stakes settings.

However, key concerns remain. The rebuttal clarifies the intended use of the framework but does not fully address issues of practical robustness. In particular, the reliance on non-independent probability estimates (e.g., LLM-generated explanations and probabilities) introduces potential circularity, limiting interpretability. Additionally, sensitivity to threshold-based abstraction and assumptions such as monotonicity raise concerns about model reliability under real-world conditions.

Overall, the rebuttal partially addressed my concerns but did not change my evaluation. While the work is original, rigorous, and promising, the unresolved issues regarding validation and robustness remain important. I therefore maintain a balanced assessment:  requiring further empirical support for practical deployment.

**Key Questions For Authors:**

1. Sensitivity Analysis: How sensitive is the Pathway Explanation Score to the specific thresholds chosen during the discretization/abstraction of continuous variables? Have you tested if small perturbations in thresholds lead to significantly different "best" pathways?
2. Computational Complexity: For very large causal graphs (e.g., complex supply chains or biological networks), does the search for the optimal causal subgraph remain computationally tractable?
3. Human Interpretability: While the paper mentions AI explainability, have there been any qualitative user studies to see if the "Pathways" identified by the algorithm align with the intuitive explanations provided by domain experts (e.g., financial analysts)?

**Limitations:**

A close examination of the paper reveals some logical restriction and simplifying assumption in its research design, analytical methodology, and interpretation of results. Although the proposed framework is mathematically sophisticated, certain methodological choices introduce potential biases and conceptual inconsistencies that may undermine the validity of the conclusions. These issues primarily concern the design of the LLM-based validation procedure, the assumptions introduced to maintain mathematical tractability, the treatment of continuous variables through binary abstraction, and the conceptual framing of causal explanations for rare events.
First, the research design exhibits a form of circular reasoning in the validation of large language model (LLM) explanations. In Section 5 and Appendix F, the authors demonstrate how the proposed mathematical framework can detect logical inconsistencies in causal explanations generated by an LLM (Claude Opus 4.5). However, the evaluation procedure requires the same LLM not only to generate the causal pathway but also to estimate the conditional probabilities along that pathway. This creates a methodological loop: if the generated explanation already contains hallucinations or structural errors, the probability distributions produced by the same model cannot be considered an independent source of evidence. Moreover, the authors explicitly acknowledge that evaluating the accuracy of these probabilities lies beyond the scope of the study. Consequently, the empirical demonstration in Section 5 is based on unverified probability estimates, raising concerns about whether the framework would remain effective when applied to real-world scenarios lacking reliable ground-truth data.
Second, the analytical framework introduces the assumption of feature monotonicity in order to maintain the mathematical validity of the main theoretical result, specifically the multivariate likelihood bound presented in Theorem 4.3. To ensure that the theorem holds, non-monotonic variables are transformed into monotonic features. The authors themselves acknowledge that this assumption does not generally hold in real-world systems. Nevertheless, they argue that “good” explanations should rely on features that do not deviate substantially from monotonicity. This reasoning presents a logical difficulty: cases that violate the monotonicity assumption are effectively dismissed as poor explanations by definition. Such reasoning resembles the fallacy of begging the question, as the validity of the explanation is implicitly defined by its conformity to the model’s assumptions. In complex systems where causal relationships are often nonlinear and non-monotonic, this restriction may significantly limit the framework’s descriptive fidelity.
Third, the practical implementation of the method relies heavily on thresholding continuous variables into binary indicators, a transformation that introduces additional theoretical complications. The causal graphical model underlying the framework depends on the Markov condition, which specifies conditional independence relations among variables. However, as the authors acknowledge, converting continuous variables into binary events can violate these independence relations. As a result, the transformed distribution may no longer satisfy the Markov property required by the causal graph. Despite recognizing this issue, the authors proceed by assuming that the resulting distribution is a reasonable approximation for analytical convenience. This assumption raises the risk of model misspecification, which could distort the computed path-explanation scores and compromise the reliability of the inferred causal structure.
Finally, the conceptual framing of causal explanation introduces a potential confusion regarding the role of counterfactual reasoning. The stated objective of the study is to identify the root cause of a particular observed rare event. However, the authors explicitly reject the use of counterfactual reasoning—the third level of causal inference in Pearl’s hierarchy—and instead rely exclusively on interventional probabilities. From the perspective of causal inference theory, explaining a singular realized event typically requires counterfactual analysis, which evaluates whether the outcome would still have occurred if a particular factor had been absent in that specific instance. The authors justify their decision by arguing that counterfactual conclusions often generalize to only a small subset of statistical units. Yet this justification appears to conflate two distinct goals: explaining a specific rare event and producing explanations that generalize to a population. If the aim is to explain a unique extreme case, generalizability may not be the primary criterion; conversely, if the objective is population-level inference, the explanation is no longer specific to that singular event. This ambiguity suggests a conceptual tension in the paper’s stated research objective.

**Strengths And Weaknesses:**

Strengths:
1. Theoretical Rigor: The study is grounded in a mathematically rigorous framework based on structural causal models and probability theory. The authors provide formal proofs for key properties of the proposed metrics, including the additivity of explanatory scores and the monotonicity of feature contributions along causal paths. These properties follow naturally from the log-likelihood decomposition implied by the Markov factorization, ensuring that the explanatory metric is theoretically principled rather than heuristic.

2. Methodological Novelty in Root Cause Analysis: Unlike conventional approaches that rely on extreme value theory or simple feature attribution, the proposed framework reconstructs the structural process through which a rare event emerges. By tracing explanatory paths across nodes in a directed acyclic graph, the method identifies both root causes and the intermediate mechanisms that propagate the anomaly, thereby providing a coherent causal narrative of system failure.

3. Practical Utility for Explainable AI: The framework offers significant value for explainable artificial intelligence, particularly in high-stakes applications. By explicitly identifying which mechanisms operate normally and which deviate, the method produces interpretable causal-path explanations that are probabilistically testable. This makes it especially useful for diagnosing failure pathways in complex AI systems and safety-critical environments.

Weaknesses:
1. High Entry Barrier for General AI Audiences: The framework relies heavily on advanced concepts from structural causal modeling, probabilistic graphical models, and information theory, resulting in a dense presentation of mathematical notation and formal derivations. While this level of formal rigor strengthens the theoretical credibility of the work, it may also limit accessibility for a broader AI readership that lacks a strong background in causal inference or statistical theory. As a consequence, readers from the general machine learning community may find it challenging to fully engage with the methodological contributions without substantial prior familiarity with these technical foundations.

2. Sensitivity Introduced by Threshold-Based Pathway Abstraction: The proposed method operationalizes causal explanations through a process referred to as pathway abstraction, in which continuous variables are converted into binary events via threshold cutting. Although this transformation facilitates the construction of interpretable causal pathways, it introduces potential sensitivity to the choice of thresholds. As discussed in the broader methodological critique, such binarization may distort the underlying probability structure and potentially violate conditional independence assumptions associated with the Markov property. Despite these implications, the main text does not systematically examine how different threshold choices might influence the resulting explanatory paths or the stability of the computed path-explanation scores, leaving an important source of model sensitivity insufficiently explored.

---

> ### Author Rebuttal · Authors · 2026-03-31
>
> We appreciate your feedback and comments. We address the questions and comments below.
>
> **W1. High Entry Barrier for General AI Audiences**
>
> Our intent was to provide a rigorous exposition, since the contribution is theoretical and mathematical density is therefore unavoidable. However, we would welcome any specific suggestions on how to make the exposition more broadly accessible without sacrificing mathematical rigor.
>
> **W2/Q1. Sensitivity introduced by threshold-based pathway abstraction**
>
> We clarify that the notion of a natural micro-realization in Definition 4.4 was introduced
> precisely to make the abstraction protocol more robust to small perturbations of the discretization
> thresholds, as noted in lines 303-325, since it averages over a range of values. Accordingly, the way we formulate causal pathway abstraction is intended to mitigate sensitivity to threshold choice.
>
> **Q2. Computational complexity for very large causal graphs**
>
> Our current contribution is theoretical, and we do not propose an algorithm for searching for the
> optimal causal pathway subgraph in large graphs. Rather, the paper establish the formal framework
> that can serve as a foundation for future algorithmic work on this search problem.
>
> **Q3. Qualitative user studies and alignment with domain experts**
>
> We did not conduct qualitative user studies with domain experts in the current work. We also clarify that the paper does not propose an algorithm for identifying or searching for causal pathways.  Rather, the goal is to formalize the notion of causal pathway and to evaluate and falsify candidate causal pathways, including those proposed by LLMs. Many pathways may sound plausible in natural language or appear convincing to a human reader, yet still fail to provide a strong causal explanation under our metric.
>
> **L1. LLM-based validation**
>
> We do not claim that the LLM provides an independent source of ground-truth probabilities. As noted in the footnote on page 8, we do not assume that these probabilities are necessarily accurate, and prior literature has highlighted this limitation [1]. Section 5 is intended as an auditing layer: under the assumption that the supplied probabilities are correct, a low explanation score indicates that the proposed causal explanation is not faithful. Passing the test, however, is not verification and does not establish correctness. As noted in the Discussion section, we do not claim that our framework already enables automated auditing of causal explanations, but rather it provides a foundation for future work in that direction.
>
> **L2. Feature monotonicity assumption**
>
> Monotonicity is an assumption, but its role in our paper is narrow: it is introduced only as an explicit assumption for the theoretical result in Theorem 4.3. Importantly, the broader framework for causal pathways and explanations does not depend on monotonicity. At the same time, our framework allows flexible choices of feature functions, and in the discussion following Theorem 4.3 (lines 281-300) we explain why features with non-monotonic relations are less appropriate for verbal explanations and why stronger causal explanations are expected when the chosen feature functions satisfy feature monotonicity.
>
> **L3. Binarization and Markov property**
>
> We agree that binary abstraction of continuous variables can alter conditional independence structure and may not preserve the Markov condition, as already noted in the paper. Our aim in introducing causal pathway abstraction is to work with an approximation, not to claim that thresholding is lossless. This is why we define a notion of accuracy in Definition 4.5. Extending the framework to continuous variables is a natural direction for future work.
>
> **L4. Counterfactual reasoning**
>
> We do not deny the importance of counterfactual reasoning in general, however counterfactuals are primarily concerned with what would have happened for a particular realized unit under an alternative hypothetical intervention, whereas our goal here is to develop a framework for causal explanations that are quantitative, falsifiable and informative beyond a single realized unit. Accordingly, our quantification focuses on explanations tied to an event of interest that generalize to a significant fraction of units and can be evaluated through explanation scores. As illustrated in example 4.6 (lines 363-381), this allows us to formulate robust causal statements such as whether an event $X \geq x$ provides a strong explanation for $Y \geq y$ in one region but not in another, which is directly testable. As further illustrated in Appendix C, counterfactual relevance does not ensure that the explanation generalizes beyond a negligible fraction of units.
>
> **References**
>
> [1] Khurram Yamin, Jingjing Tang, Santiago Cortes-Gomez, Amit Sharma, Eric Horvitz, and Bryan Wilder. Do LLMs act like rational agents? Measuring belief coherence in probabilistic decision making. arXiv preprint arXiv:2602.06286, 2026.

---

> > ### Author Rebuttal · Reviewer_uvjU · 2026-04-03
> >
> > Q3.
> > The response mainly reframes the method as a conditional, internal-consistency check under correct probabilities, but does not address the core concern: how the framework behaves or remains useful when those probabilities are biased or not independently obtained. As such, the practical implications of the circularity issue remain unresolved.
> >
> > L1.
> > The proposed falsifiability relies on observational probabilities satisfying (or violating) a factorization implied by the pathway DAG. However, when these probabilities are not independently obtained (e.g., generated by the same LLM), violations or satisfactions of the factorization cannot be unambiguously attributed to the pathway itself. As a result, the falsifiability signal becomes confounded with errors in probability estimation, limiting its practical interpretability.

---

> > > ### Author Response · Authors · 2026-04-07
> > >
> > > We thank you for this follow-up. In the experimental setting where both the pathway and the required probabilities are obtained from the LLM, the resulting score should be interpreted primarily as an internal consistency audit of the model's proposed causal explanation. The reason is that both the hypothetical causal pathway and the probability estimates come from the same source, so the test evaluates whether the proposed pathway is consistent with the model's believed probabilities.
> > >
> > > However, we wish to clarify that our auditing framework is not inherently tied to purely LLM-generated probabilities. As indicated in the footnote on page 8, the LLM could directly estimate probabilities from data. We define pathways relative to a context, and that context can be understood as determining a statistical population from which the relevant conditional probabilities may in principle be estimated. Thus, the conditional probabilities of each event given its parent events in the pathway could be estimated from external observational data rather than from LLM guesses alone. In that setting, the framework evaluates the proposed pathway as a causal hypothesis against external observational data, and failure of the induced factorization is more directly interpretable as evidence against the proposed causal pathway. One possible direction for obtaining more reliable probability estimates is to enable access to external data for estimating the required conditional probabilities, for example through retrieval-augmented setups. We will revise the paper to clarify this distinction more explicitly and to discuss an illustrative retrieval-augmented setup for obtaining the required conditional probabilities from external observational data.

---

### Official Review · Reviewer_75S5 · 2026-03-13

**Soundness:** 3
**Presentation:** 3
**Significance:** 3
**Originality:** 3
**Overall Recommendation:** 5
**Confidence:** 3

**Summary:**

This paper formalizes the causal pathway connecting root causes to target rare events in structural equation models. Based on the explanation score, the authors define cluster explanation score and pathway explanation score to quantify the degree to which a set of root causes explain observed events. They also introduce pathway abstraction from fine-grained causal models via binarization and feature monotonicity, and discuss testable implications for falsifying causal explanations. The proposal is illustrated with examples on continuous and discrete variables, and applied to evaluate an LLM-generated causal pathway.

**Compliance With Llm Reviewing Policy:**

Affirmed.

**Final Justification:**

This is a well-written theory paper that formalizes causal pathways for rare events in a principled and original way. The framework is technically sound, clearly presented, and the examples effectively illustrate the key concepts. The rebuttal addressed my main concerns, particularly the distinction between cluster and pathway explanation scores.

Thus, I have updated my score from 4 to 5, which reflects confidence in the paper's contribution to the causal inference literature.

**Key Questions For Authors:**

- Is there a one-to-one correspondence between a cluster and a pathway, or can one cluster correspond to multiple pathways?
- Can multiple different pathways have the same cluster explanation score? If so, how should we explain it?
- In Figure 1, under the pathway explanation score, $B_1$, $B_3$, and $B_4$ are root causes, whereas under the cluster score only $B_1$ and $B_3$ are. Would adding $B_2$ to the root causes increase the cluster explanation score?

**Limitations:**

The framework assumes a known causal DAG and Markov condition; in practice, the full graph is often unknown. Pathway abstraction relies on feature monotonicity, which may not hold generally.

**Strengths And Weaknesses:**

Strengths:
- The motivation is clear and the problem is very interesting. Previous root cause analysis usually identifies the cause, but does not determine the mechanism of its propagation. Formalizing the approach of connecting verbal explanations and detailed causal modeling is meaningful.
- The paper proposes a formal definition of causal pathway and discusses its testable implications.
- The additivity of the cluster explanation score is a useful property that avoids the need for Shapley values and makes attribution straightforward.
- The difference between cluster and pathway explanation scores is well motivated.
- The examples and case studies illustrate the concepts concretely.



Weaknesses:
- The testable implications could be stronger. It is currently unclear to what extent observational probabilities alone can falsify causal pathways (especially when the full causal graph is unknown). The paper acknowledges that automated testing is not yet achievable, but the scope of falsifiability could be discussed more clearly.
- The LLM internal consistency check in Section 5 is ad-hoc. LLM outputs may be inconsistent with their internal beliefs across queries.
- The relationship between cluster explanation score and pathway explanation score seems to be complementary. It would be hard to say which one should be preferred in general or how to make a choice between them when they disagree.

---

> ### Author Rebuttal · Authors · 2026-03-31
>
> We sincerely appreciate your constructive feedback and insightful comments. We address the questions and comments below.
>
> **W1. Falsifiability from observational probabilities.**
>
> Pathway abstraction is intended to enable falsifiability without requiring the full underlying causal graph. Up to the approximation error determined by the abstraction accuracy, the joint probability distribution of the pathway events should approximately factorize according to the pathway DAG itself, and not only relative to the full causal graph. Hence, observational probabilities can provide a meaningful falsifiability test: if the induced factorization fails, then the proposed pathway cannot be a faithful approximate abstraction. This is analogous to the causal Markov condition for a causal DAG: it provides a necessary sanity check, but not a sufficient verification criterion. Thus, observational probabilities can falsify pathways incompatible with the probabilistic structure implied by the abstraction, without proving correctness.
>
> **W2. LLM internal consistency check**
>
> As clarified in the footnote on page 8, we do not assume that the probabilities are necessarily accurate, and this limitation has also been noted in prior work [1]. Rather than treating these probabilities as verified, our goal is to introduce an auditing layer: under the assumption that they are correct, a low score is evidence that the proposed causal explanation is not faithful. In other words, if the LLM’s causal explanation were correct, it should at least satisfy this criterion. Passing the test, however, should not be interpreted as verification. We also emphasize that the paper is not about assessing the quality of state-of-the-art LLMs. Rather, it formalizes a notion of internal consistency that can serve as an additional audit for proposed causal explanations.
>
> **W3. Cluster vs. pathway explanation score.**
>
> The pathway explanation score measures whether the root causes are sufficient to explain the *target*, while the cluster explanation score measures whether they sufficiently explain the *cluster*. To see the difference, consider the pathway $X_1 \to X_2 \to X_3$, where $X_1$ is the unique root cause. Assume $P(X_1=1)=10^{-15}$ and $X_2=X_1$. Further, $P(X_3=1|X_2=0)=10^{-3}$ and  $P(X_3=1|X_2=1)=10^{-5}$, that is, the event $X_1=1$ renders $X_3=1$ even *less* likely. The cluster explanation score reads $1- \frac{\log (10^{-5}) }{\log (10^{-20})} = 1- 5/20= 3/4$, while the pathway explanation score is  $1- \frac{\log (10^{-5})}{\log ((1- 10^{-15}) 10^{-3} + 10^{-15} 10^{-5})} \approx 1- 5/3 <0$.  For the very unlikely joint event $(X_1=1,X_2=1, X_3=1)$, $X_1=1$ is a good explanation since it renders that joint event significantly more likely. However, the target event $X_3=1$ is not itself that unlikely. Therefore, $X_1=1$ is a poor explanation of the target, since it makes the target even less likely.
>
> **Q1. One-to-one correspondence between cluster and pathway?**
>
> Not in general. A single cluster may give rise to multiple pathways, depending on the choice of target node. A causal pathway refines a cluster by specifying a target, which is why we define the target score and the pathway explanation score, whereas the cluster score evaluates the explanatory quality of the cluster without reference to a specific target. Once a target is fixed, only the subgraph upstream of that target is relevant to the corresponding pathway.
>
> **Q2. Can multiple different pathways have the same cluster explanation score?**
>
> Yes, multiple pathways may have the same cluster explanation score if they are induced by the same cluster graph and the same set of root causes, but differ only in the choice of target node. The reason is that the cluster explanation score assesses the explanatory quality of the cluster as a whole, without reference to a specific target. Differences between such pathways are instead reflected in the target score and the pathway explanation score.
>
> **Q3. In Figure 1, would adding $B_2$ to the root causes increase the cluster explanation score?**
>
> Adding $B_2$ to the set of root causes does not increase either the pathway explanation score or the cluster explanation score, since $P(b_2^1|b_1^1)=1$. Intuitively, $B_2$ is a propagating node that occurs deterministically whenever the preceding event occurs, and is therefore not treated as an additional root cause. More generally, for both causal pathways and clusters, one may consider multiple candidate sets of root causes. The preferred set is the one that has the highest explanation score among those that satisfy the quality of explanation criteria stated in lines 202-233.
>
> **References**
>
> [1] Khurram Yamin, Jingjing Tang, Santiago Cortes-Gomez, Amit Sharma, Eric Horvitz, and Bryan
> Wilder. Do LLMs act like rational agents? Measuring belief coherence in probabilistic decision
> making. arXiv preprint arXiv:2602.06286, 2026.

---

> > ### Author Rebuttal · Reviewer_75S5 · 2026-04-04
> >
> > I appreciate the author's detailed response. My questions have been largely addressed, and I particularly appreciate the clarification on the distinction between cluster and pathway explanation scores.
> >
> > A small suggestion for revision is to briefly indicate near Lemma 4.2 and Theorem 4.3 that these results are directly related to the testable implications of causal pathways, as this connection is not clear from the current Section 4.
> >
> > I have updated my score accordingly.

---

> > > ### Author Response · Authors · 2026-04-07
> > >
> > > We thank you for the thoughtful follow-up and for updating your score. We are glad that the clarification on the distinction between cluster and pathway explanation scores was helpful. We also thank you for the suggestion regarding Lemma 4.2 and Theorem 4.3. We will revisit the exposition in Section 4 accordingly.

---

### Official Review · Reviewer_W1G4 · 2026-03-13

**Soundness:** 4
**Presentation:** 4
**Significance:** 3
**Originality:** 3
**Overall Recommendation:** 5
**Confidence:** 3

**Summary:**

The authors propose a framework to formalize causal explanations of rare events. This framework bridges verbal explanations of such phenomena, which often rely on chains of rare events, with causal modeling formalisms. To achieve this, the authors introduce a formal notion of causal pathways and build on the idea of explanation scores, applying and extending them to explain clusters of events. They also introduce an abstraction of causal structure that reduces the set of nodes in a DAG by representing events of interest as binary variables and by ‘featurizing’ variables such that their definitions incorporate their original mechanisms. They use these formalisms together to propose criteria to measure the quality of explanations as well as external and internal consistency checks for causal explanations of rare events based on pathways that can be applied to, e.g., LLMs.

**Compliance With Llm Reviewing Policy:**

Affirmed.

**Final Justification:**

I thank the authors for their clarifications and responsiveness during the rebuttal period. The rebuttal and additional comparison to be incorporated in the final version of the paper have addressed my concerns. I thus maintain my positive score of 5 and advocate for the paper's acceptance.

**Key Questions For Authors:**

- [Question 1] - Are there other methods for explanations of rare events for which a direct comparison (perhaps through an example) could further highlight the benefits of this framework compared to other approaches?
- [Question 2] - I have a clarification question about Example 3.6. The stated target event is $b_4^1$ ($B_4 = 1$), but in lines 197-199, the discussion states that “our surprise about the node $B_4$ is moderate, since it propagates the signal 1 in 1 out of 100 cases, which is not that rare compared to how rare the target event is.” Can the authors further clarify how the event $b_4^1$ is distinct from $B_4$ propagating the signal 1?
- [Note] - The example in Figure 2 with isolines says that for outliers above 2, the explanation score reaches at least 80%. If I am reading the plot correctly, I think this should say 72%. Or alternatively, it should say the outliers are above 3.

**Limitations:**

yes

**Strengths And Weaknesses:**

The paper is well written and clearly structured. The exposition is precise, well-situated in the related literature for its core ideas, and builds in a logical manner that makes reading the paper a pleasant experience. The formalisms appear technically sound, and in my view the paper has the potential to be a significant contribution. Many of the key insights are to my knowledge original, and for those that are not, their original source is cited. The related work appears to be sufficient, but I think the paper could benefit from some additional discussion of existing causal explanation techniques, perhaps through an example that highlights concretely the benefits of this framework compared to other approaches.

---

> ### Author Rebuttal · Authors · 2026-03-31
>
> We sincerely appreciate your positive feedback and thoughtful comments. We address the questions and comments below.
>
> **Q1. Direct comparison to other methods for explaining rare events.**
>
> There is a rich literature on path-specific effects, e.g., [1]. However such approaches typically quantify contributions in terms of differences in conditional means, as is also common in feature-relevance methods [2, 3]. In contrast, following [4, 5], our framework quantifies contributions on the logarithmic scale, which places strong weight on those causes of rare events that are rare themselves. For instance, in many cases a rare cause may contribute strongly to the occurrence of a rare event when present, yet only have a limited average mean effect under mean-based measures and thus be underemphasized, whereas our framework assigns it substantially greater explanatory weight.
>
> **Q2. Clarification about Example 3.6.**
>
> We thank you for pointing this out. Our wording here may have been misleading. The target event is $B_4=1$ (denoted $b_4^1$), which is indeed rare in absolute terms. The statement that $B_4$  "propagates the signal 1 in 1 out of 100 cases" was intended to refer to the conditional probability leading to the target event, namely $P(b_4^1|b_3^1)=1/100$. Thus, the intended meaning was that, conditional on $B_3=1$, the event $B_4=1$ is not particularly rare relative to the other conditional probabilities appearing in the pathway, even though it is rare marginally. We will revise the wording to make this distinction clearer.
>
> **Q3. Isolines in Figure 2.**
>
> We thank you for catching this typo. It should read ‘outliers above 3'. We will correct this in a revised version of the paper.
>
> **References**
>
> [1] C. Avin, I. Shpitser, and J. Pearl. Identifiability of path-specific effects. In Proceedings of the
> International Joint Conference in Artificial Intelligence, pages 357–363, Edinburgh, Scotland, 2005.
>
> [2] S. Lundberg and S. Lee. A unified approach to interpreting model predictions. In I. Guyon, U. V.
> Luxburg, S. Bengio, H. Wallach, R. Fergus, S. Vishwanathan, and R. Garnett, editors, Advances
> in Neural Information Processing Systems 30, pages 4765–4774. Curran Associates, Inc., 2017.
>
> [3] S. Kleinberg. Causal inference with rare events in large-scale time-series data. In Proceedings of the Twenty-Third International Joint Conference on Artificial Intelligence, IJCAI ’13, page 1444–1450.
> AAAI Press, 2013.
>
> [4] Kailash Budhathoki, Lenon Minorics, Patrick Bloebaum, and Dominik Janzing. Causal
> structure-based root cause analysis of outliers. In Kamalika Chaudhuri, Stefanie Jegelka, Le Song,
> Csaba Szepesvari, Gang Niu, and Sivan Sabato, editors, Proceedings of the 39th International
> Conference on Machine Learning, volume 162 of Proceedings of Machine Learning Research, pages 2357–2369. PMLR, 17–23 Jul 2022.
>
> [5] Michael Oesterle, Patrick Bl¨obaum, Atalanti Mastakouri, and pages= year=2025 organization=
> Kirschbaum Elke, booktitle=Proceedings of Conference on Causal Learning and Reasoning
> (CLeaR). Beyond single-feature importance with icecream.

---

> > ### Author Rebuttal · Reviewer_W1G4 · 2026-04-03
> >
> > The author rebuttal largely addressed my questions, but I think the work could still benefit from a concrete example comparison to other explanation methods, for example demonstrating this aforementioned underemphasis that a mean-based measure might place on a rare cause compared to the method in this work. That said, I do not see this as a critical issue with the paper and thus maintain my score.

---

> > > ### Author Response · Authors · 2026-04-07
> > >
> > > We thank you for this follow-up. To provide a concrete illustration with mean-based attribution, consider the logical AND gate $Y=X \wedge N$ with $P(X=1)=10^{-9}$ and $P(N=1)=10^{-1}$, so that $P(Y=1)=10^{-10}$. With logarithmic scaling (following explanation scores), the contribution of $X$ reads $1 - \frac{1}{10}=90$%, in other words its contribution is strong because it increases the probability of the output event by a large factor. By contrast, a mean-based contribution such as $\frac{\mathbb{E}[Y|do(X=1)]-\mathbb{E}[Y]}{1-\mathbb{E}[Y]}$ is approximately $10^{-1}$, which is much smaller. If we instead consider Shapley-based contributions, the contribution of $X$ is $\frac{\mathbb{E}[Y|X=1] - \mathbb{E}[Y] + \mathbb{E}[Y|X=1,N=1]- \mathbb{E}[Y|N=1]}{2(1-\mathbb{E}[Y])} \approx 0.55$.
> > >
> > > This gives a stronger contribution than the plain mean-based score, but it still remains substantially below the logarithmic score of $0.9$, since the marginal terms are still tied to the absolute change in the mean. Since we follow the arguments in [1, 2] that a substantially more rare factor should get significantly more attention, we prefer the logarithmic contribution analysis. Another issue with Shapley-based contributions is that they require knowing the structural causal model and are not invariant under redefining the noise: if we replace $N$ by two variables $N_1, N_2$, with $Y = X \wedge N_1 \wedge N_2$ and $P(N_1=1, N_2=1)=P(N=1)=10^{-1}$, then the Shapley contribution of $X$ changes. We will include a version of this example in the revised paper.
> > >
> > >
> > > **References**
> > >
> > >
> > > [1] Oesterle, M., Blobaum, P., Mastakouri, A. A., and Kirschbaum, E. Beyond single-feature importance with
> > > icecream. In Huang, B. and Drton, M. (eds.), Proceedings of the Fourth Conference on Causal Learning and Reasoning, volume 275 of Proceedings of Machine Learning Research, pp. 359–389. PMLR, 07–09 May 2025. URL https://proceedings.mlr.press/v275/oesterle25a.html.
> > >
> > > [2] Kailash Budhathoki, Lenon Minorics, Patrick Bloebaum, and Dominik Janzing. Causal structure-based root cause analysis of outliers. In Kamalika Chaudhuri, Stefanie Jegelka, Le Song, Csaba Szepesvari, Gang Niu, and Sivan Sabato, editors, Proceedings of the 39th International Conference on Machine Learning, volume 162 of Proceedings of Machine Learning Research, pages 2357–2369. PMLR, 17–23 Jul 2022.

---

### Official Review · Reviewer_NBqu · 2026-03-13

**Soundness:** 4
**Presentation:** 4
**Significance:** 3
**Originality:** 4
**Overall Recommendation:** 5
**Confidence:** 4

**Summary:**

The authors present a new formal definition for the causal pathways of rare events. Root cause analysis of rare events typically identifies a set of root causes, but does not capture the causal mechanism that connects them to the event. The present paper is a theory paper.

**Compliance With Llm Reviewing Policy:**

Affirmed.

**Key Questions For Authors:**

N/A (Thanks!)

**Limitations:**

Yes.

**Strengths And Weaknesses:**

(+) Relevant conceptual improvement. Establishing a formal definition that is consistent with the existing literature is crucial.
(+) Important properties and results alongside examples are established while arriving at the definitions. Very much inline with similar works by for instance Y. Halpern or S. Beckers that have positively shaped the literature.
(+) Clear presentation, discussion and disclaimers on limitations/open work.

---

> ### Author Rebuttal · Authors · 2026-03-31
>
> We sincerely thank you for the positive feedback. We are pleased that you found the formalization well grounded in the existing literature, and that the properties, examples, and discussion of limitations came through clearly.

---

### Decision · Program_Chairs · 2026-04-30

**Decision:**

Accept (regular)

**Comment:**

This is a well‑written theoretical paper that presents a formal framework for representing and evaluating causal‑pathway‑based explanations of rare events. All reviewers agree that the contributions are original, technically rigorous, and potentially useful for root cause analysis and explanatory reasoning more broadly.

The authors responded well to the reviewers’ questions and concerns in their rebuttal, particularly with respect to the comparison of different quantification measures, the approximate nature of pathway abstraction, the application to consistency checking of relevant LLM outputs, and the mathematical density of the presentation. Most reviewers are fully satisfied with the responses, and all reviewers rate the paper as acceptable, including three 'accept' and one 'weak accept'. My own assessment aligns with this consensus.